# Phase interface engineering enables state-of-the-art half-Heusler thermoelectrics

Yihua Zhang[1,2,7], Guyang Peng[2,7], Shuankui Li[3,7], Haijun Wu [2] ✉, Kaidong Chen[1], Jiandong Wang[2], Zhihao Zhao[2], Tu Lyu[1], Yuan Yu [4], Chaohua Zhang [1], Yang Zhang[5,6], Chuansheng Ma[6], Shengwu Guo[2], Xiangdong Ding[2] ✉, Jun Sun [2], Fusheng Liu[1] ✉ & Lipeng Hu [1] ✉

In thermoelectric, phase interface engineering proves effective in reducing the lattice thermal conductivity via interface scattering and amplifying the density-of-states effective mass by energy filtering. However, the indiscriminate introduction of phase interfaces inevitably leads to diminished carrier mobility. Moreover, relying on a singular energy barrier is insufficient for comprehensive filtration of low-energy carriers throughout the entire temperature range. Addressing these challenges, we advocate the establishment of a composite phase interface using atomic layer deposition (ALD) technology. This design aims to effectively decouple the interrelated thermoelectric parameters in ZrNiSn. The engineered coherent dual-interface energy barriers substantially enhance the density-of-states effective mass across the entire temperature spectrum while preser carrier mobility. Simultaneously, the strong interface scattering on phonons is crucial for curtailing lattice thermal conductivity. Consequently, a 40-cycles $TiO_2$ coating on $ZrNi_{1.03}Sn_{0.99}Sb_{0.01}$ achieves an unprecedented $zT$ value of 1.3 at 873 K. These findings deepen the understanding of coherent composite-phase interface engineering.

The usage of thermoelectric materials holds great promise for the direct conversion of waste heat into useful electricity, making them valuable candidates for energy conversion applications and sustainable energy development. The efficiency of thermoelectric device depends significantly on the material's dimensionless Fig of merit, denoted as $zT = \sigma\alpha^2 T/\kappa$, where $\alpha$, $\sigma$, $\kappa$, and $T$ are the Seebeck coefficient, electrical conductivity, total thermal conductivity (including lattice contribution $\kappa_L$ and carrier contribution $\kappa_e$), and the absolute temperature, respectively[1]. A more detailed evaluation of $zT$ involves the quality factor $\beta \propto \mu_H m^{*3/2}/\kappa_L$, which reveals the utmost importance of

maintaining high carrier mobility ($\mu_H$) in materials with low $\kappa_L$[2]. Recent efforts toward high $zT$ are in line with the quality factor, which boost the density-of-states effective mass ($m^*$) by band structure manipulation[3–7], as well as reducing $\kappa_L$ via multiscale microstructures[8–16].

As an emerging paradigm-shifting strategy, phase interface engineering plays a pivotal role in optimizing thermoelectric transport parameters. This strategy addresses two key aspects: enhancing $m^*$ via energy filtering effect and shrinking $\kappa_L$ through dense phonon scattering at high-density interfaces[17–22]. However, leveraging phase interface engineering to achieve superior thermoelectric performance

---

[1]College of Materials Science and Engineering, Shenzhen Key Laboratory of Special Functional Materials, Guangdong Research Center for Interfacial Engineering of Functional Materials, Guangdong Provincial Key Laboratory of Deep Earth Sciences and Geothermal Energy Exploitation and Utilization, Institute of Deep Earth Sciences and Green Energy, Shenzhen University, Shenzhen 518060, China. [2]State Key Laboratory for Mechanical Behavior of Materials, Xi'an Jiaotong University, Xi'an 710049, China. [3]School of Physics and Materials Science, Guangzhou University, Guangzhou 510006, China. [4]Institute of Physics (IA), RWTH Aachen University, Sommerfeldstraße 14, 52074 Aachen, Germany. [5]Electronic Materials Research Laboratory (Key Lab of Education Ministry) and School of Electronic Science and Engineering, Xi'an Jiaotong University, Xi'an 710049, China. [6]Instrumental Analysis Center of Xi'an Jiaotong University, Xi'an Jiaotong University, Xi'an 710049, China. [7]These authors contributed equally: Yihua Zhang, Guyang Peng, Shuankui Li. ✉e-mail: wuhaijunnavy@xjtu.edu.cn; dingxd@xjtu.edu.cn; fsliu@szu.edu.cn; hulipeng@szu.edu.cn

faces two significant challenges. Firstly, interface potential barriers between nanoinclusions and the matrix effectively filter out low-energy charge carriers, thereby augmenting the density of states (DOS) proximal to the Fermi level. Yet, this enhancement of $m^*$ tends to diminish at elevated temperatures due to the limitations imposed by a single energy potential barrier ($\Delta E$)[23–25]. Secondly, arbitrarily introduced phase interfaces can scatter both phonons and carriers, leading to a detrimental effect on carrier mobility ($\mu_H$) and hence the $zT$[26,27]. For example, the introduction of $Al_2O_3$ into ZrNiSn or the creation of amorphous ZrNiSn have both successfully led to a significant reduction in $\kappa_L$. However, these approaches did not effectively enhance $PF$ due to the decrease in $\mu_H$. Consequently, the coupling between electronic and phonon transport properties has limited the potential for improving the figure of merit $zT$[28,29]. To date, achieving a significant enhancement in $\mu_H m^{*3/2}/\kappa_L$ over the entire temperature range of thermoelectric materials remains a challenging and elusive goal[30]. Hence, a judicious and rational phase interface design is crucial for implementing phase interface engineering in thermoelectrics.

In this context, the construction of a coherently composite-phase interface is proposed as a promising strategy for the synergistic optimization of $\mu_H m^{*3/2}/\kappa_L$. The connotation of a composite phase interface contains two fundamental principles. Initially, theoretical considerations suggest that the optimal energy barrier should lie between 1 to 10 $k_B T$ ($k_B$ and $T$ stand for Boltzmann's constant and absolute temperature)[31], a range that varies with temperature. To ensure a consistent enhancement of $m^*$ across varying temperatures, the establishment of multiple energy barriers is crucial, ensuring the filtration of low-energy electrons throughout the entire temperature spectrum[23]. Additionally, an ideal phase interface should scatter phonons more effectively than electrons. Analysis of the phase interface structure reveals that both coherent and incoherent interfacial structures lead to an obvious decrease in $\kappa_L$. However, materials with coherent interfaces exhibit markedly higher $\mu_H$ than those with incoherent interfaces[32,33]. Despite these insights, comprehensive studies that concurrently address these aspects remain scarce. The construction of coherent phase interfaces harboring multiple $\Delta E$ is pivotal in disentangling the intricately linked thermoelectric parameters[18,23,34]. However, the primary challenge lies in the systematic engineering of these coherent composite phase interfaces featuring multiple $\Delta E$ in experimental setups to optimize the $zT$. Historically, methods such as ball milling[35], hydrothermal synthesis[36], self-precipitation[37,38], and melt spinning[39] have been prevalent for introducing secondary phases into the matrix. Regrettably, these methods fall short in precisely constructing coherent phase interfaces with multiple $\Delta E$. To surmount this challenge, employing atomic layer deposition (ALD) for the manipulation of coherent composite-phase interfaces emerges as a strategically effective pathway. ALD, renowned for its self-limiting and self-saturating characteristics, is ideal for the layer-by-layer growth of 2D films. Crucially, the unique capability of ALD technology to uniformly coat diverse 3D bulk materials enables meticulous control over the thickness of the coating layer[40]. At elevated sintering temperatures, the amorphous nature of these coating enhances chemical reactivity, potentially initiating in-situ chemical reactions between the substrate and the ALD-coated layer. In this study, we present a unique methodology for constructing coherent composite-phase interfaces with multiple $\Delta E$ through the in-situ reflection between the matrix and the ALD coating. For this purpose, the half-Heusler (HH) ZrNiSn alloy was chosen as an exemplary template for implementing phase interface engineering due to its high sintering temperature. We employed an ALD coating of amorphous $TiO_2$, chosen for its high chemical reactivity, as a precursor layer. The coherent composite-phase interfaces within the $n$-type $ZrNi_{1.03}Sn_{0.99}Sb_{0.01}$ matrix (referred to ZNSS) is thereafter formed after high-temperature sintering. The initially amorphous $TiO_2$ coating undergoes a comprehensive transformation, being replaced by a Ti-ZNSS layer (aka $Zr_{1-x}Ti_xNi_{1.03}Sn_{0.99}Sb_{0.01}$) along

with uniformly dispersed $ZrO_2$ nanoparticles, as depicted in the schematic diagram in Fig. 1a.

The in-situ formation of the composite phase interfaces, consisting of the Ti-ZNSS layer and $ZrO_2$ nanoparticles, impeccably orchestrates the simultaneous optimization of electrical and thermal transport through two synergistic mechanisms. Firstly, the augmentation of the energy filtering effect on $m^*$ facilitated by various $\Delta E$ strategically created between the matrix and composite phase, as illustrated in the schematic diagram in Fig. 1c. Secondly, the introduction of coherent phase interfaces substantially reduces the $\kappa_L$ while maintaining $\mu_H$ unchanged, as depicted in the schematic diagram in Fig. 1d. Furthermore, we are pleasantly surprised to discover that there are a significant number of twins within the $ZrO_2$ nanoparticles. This twin structure provides an additional scattering center for phonon transport at the nanoscale. Consequently, the state-of-the-art $zT$ of 1.3 is attained in the ZNSS sample coated with 40 cycles of $TiO_2$ (i.e., $TiO_2$ = 3.2 nm) (Fig. 1b). These findings underscore the pivotal role of coherent composite-phase interface engineering in achieving cutting-edge $zT$ in ZrNiSn-based and other thermoelectric materials.

## Results

### Design principle

Designing composite phase structures with coherent phase interfaces and multiple $\Delta E$ is aimed at enhancing $m^*$ and phonon scattering while minimally impacting the $\mu_H$. Given the necessity for high-temperature sintering to facilitate the interface chemical reaction between matrix and amorphous ALD coating layer, ZrNiSn-based materials, known for their high melting point, are selected as an ideal platform for constructing the composite phase interfaces. The initial step involves the synthesis of a high purity ZNSS matrix. To this end, the ZNSS matrix is fabricated using levitation melting, a technique chosen for its efficiency in producing high-purity materials. The room temperature powder X-ray diffraction (PXRD) patterns, as illustrated in Fig S1, reveal the diffraction peaks of the ZNSS matrix that are exclusively indexed to the cubic MgAgAs-type crystal structure without any detectable second phases. A critical prerequisite for the construction of coherent composite-phase interfaces is the preparation of ZNSS powders uniformly coated with amorphous $TiO_2$ layers. The dynamic formation process of depositing ultrathin $TiO_2$ layers onto the ZNSS powders through ALD is meticulously captured in Fig. 2a. During the ALD process, conducted at 423 K, tetrakis (dimethylamido) titanium (TDMAT) and $H_2O$ serve as precursors for $TiO_2$ deposition[41,42]. Notably, the PXRD pattern of the ALD-coated samples (Fig. S2) reveals the absence of crystalline $TiO_2$ diffraction peaks, as the $TiO_2$ layer obtained is actually amorphous and less than 10 nm, which is beyond the detection limit of XRD. Further validation of the successful coating of the ultrathin $TiO_2$ layer on the surface of ZNSS powder is obtained through X-ray photoelectron spectroscopy (XPS). The XPS analysis, particularly for the coated 40 cycles $TiO_2$ sample, demonstrates a distinct Ti 2p peak, unequivocally indicating the presence of the amorphous $TiO_2$ layers on the ZNSS surfaces (Fig S3)[43].

The ALD boasts significant advantages, particularly in its capability to homogeneously coat complex 3D structures and precisely control phase thickness. The quality of the ALD coating layer is validated through high-resolution transmission electron microscopy (HRTEM) images of a representative coated $TiO_2$ powder coated for 40 cycles, as illustrated in Fig. 2b. It is evident that $TiO_2$ with 40 cycles is uniformly enveloped by an ultrathin amorphous $TiO_2$ layer, approximately 3 nm thick. The calculated growth rate of $TiO_2$, is around 0.8 Å per cycle, which aligns with previous literatures[40,44]. The resulting samples, differentiated by the number of ALD-coated $TiO_2$ layers, are designated as uncoated, $TiO_2$ = 0.8 nm (10 cycles), $TiO_2$ = 1.6 nm (20 cycles), $TiO_2$ = 3.2 nm (40 cycles), $TiO_2$ = 4.8 nm (60 cycles) and $TiO_2$ = 6.4 nm (80 cycles). Such meticulous characterizations underscore the successful deposition of high-quality, ultrathin amorphous

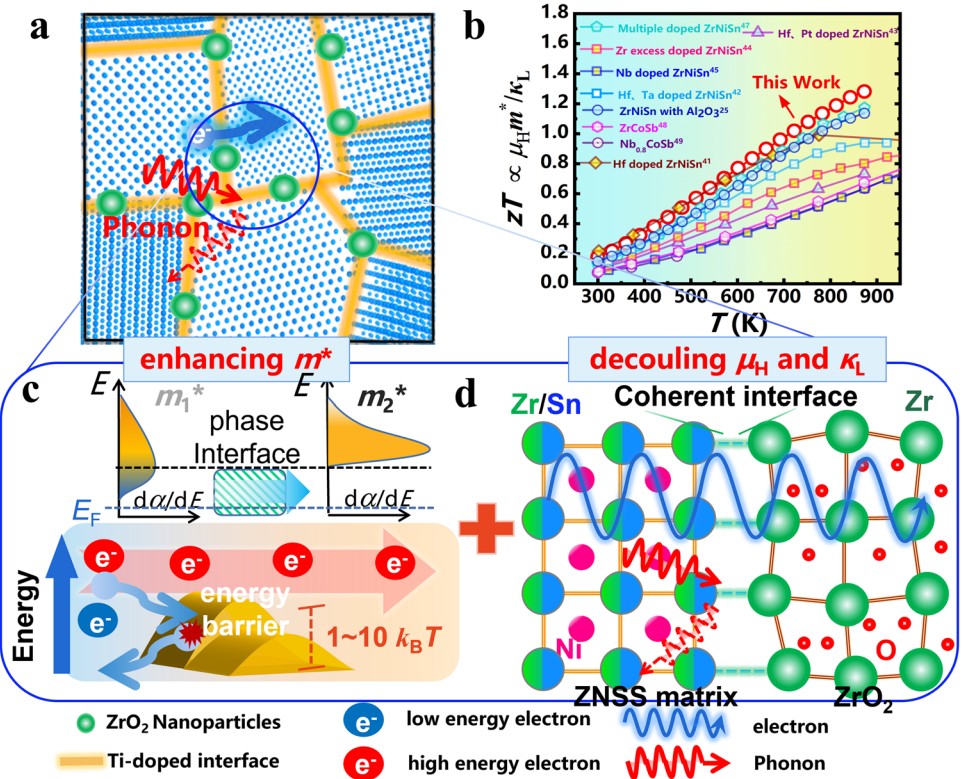

**Fig. 1 | Schematic diagram of the decoupled thermoelectric properties of the working action of the coherent composite-phase. a** Schematic diagram of electron and phonon transport in the multi-scale phase interfaces. **b** Temperature-dependent $zT$ for the $TiO_2 = 3.2$ nm sample in this work, compared with those of other high-$zT$ HH TE materials[25,49,59–66]. **c** Schematic diagram of energy filtering effect. **d** Schematic diagram of electron and phonon transport in the coherent phase interfaces.

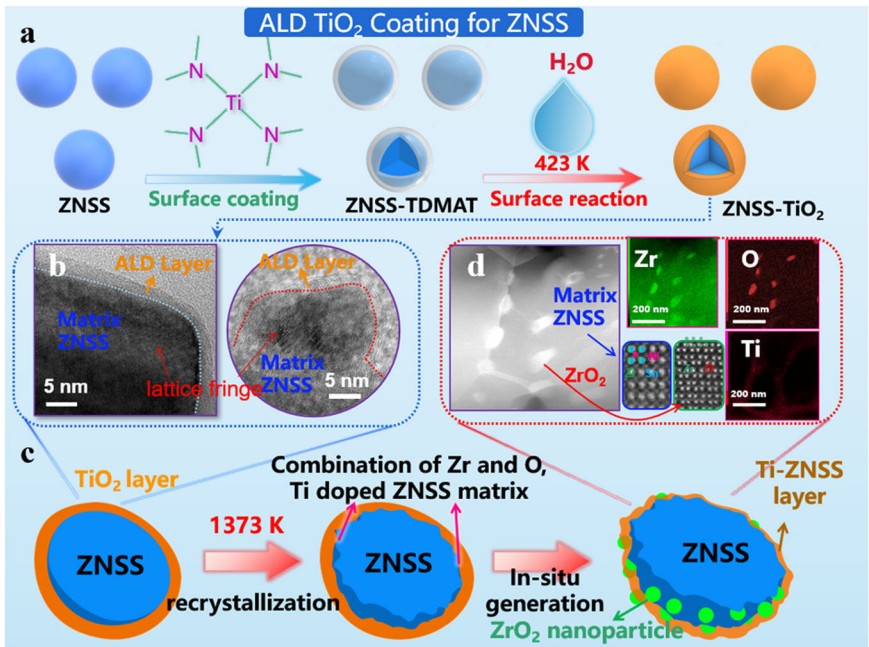

**Fig. 2 | ALD synthesis technology route. a** Schematic diagram illustrating the synthesis of the $TiO_2$ layer through ALD coating. **b** HRTEM images of ZNSS powders coated with 40 cycles of $TiO_2$ ($TiO_2 = 3.2$ nm). **c** Schematic representation of the interfacial reaction between the ZNSS matrix and ALD coating. **d** ABF images and corrsponding (Zr, O, Ti element) EDS mapping of the bulk ZNSS sample coated with 40 cycles of $TiO_2$.

$TiO_2$ layers on the surface of ZNSS powder, laying a robust foundation for the construction of composite phase interfaces.

The choice of amorphous $TiO_2$ for the ALD coating is strategic, considering high chemical activity and susceptibility to oxygen deficiency at elevated temperatures, which are conducive to fostering the desired interfacial reactions. To expedite the chemical reaction at the interface between the $TiO_2$ coating layer and ZNSS powder, a densification process is undertaken using spark plasma sintering (SPS)

at 1373 K for 10 min. It is worth noting that no significant second phase was observed in the XRD patterns after sintering (Figs. S4 and S5). A comprehensive schematic depicting this interface chemical reaction is presented in Fig. 2c. In the initial stage, the O atoms within the amorphous $TiO_2$ layer bond with the Zr atoms from the matrix, resulting in the formation of $ZrO_2$ nanoparticles. Simultaneously, the residual Ti atoms occupy vacant Zr sites, creating a Ti-ZNSS layer. This sophisticated interfacial reaction mechanism is further substantiated by scanning transmission electron microscopy (STEM) images, including both annular bright field (ABF and corresponding EDS mapping) and atomically-resolved high angle annular dark field (HAADF) images, as shown in Fig. 2d. There images validate the uniform distribution of nanoparticles at the grain boundaries of the substrate, consistent perfectly with the hypothesized interface design. This intricate interplay at the nanoscale level highlights the ingenuity and precision of the phase interface engineering approach employed in this study.

## Microscopic characterization of composite phase interfaces

In order to achieve a more precise elucidation of the reaction product of the SPSed $TiO_2 = 3.2$ nm sample, we employed scanning electron microscope (SEM) and electron probe micro analysis (EPMA) to unveil the intricate composite phase compositions. Firstly, the X-ray energy dispersive spectrum (EDS) mapping and spot scanning results (Tables S1–S4) disclosed a noteworthy observation: Ti atoms did not enter into the ZNSS matrix. Instead, they replaced the pristine amorphous $TiO_2$ layer, culminating in a uniform Ti-ZNSS layer, formally identified as $Zr_{1-x}Ti_xNi_{1.03}Sn_{0.99}Sb_{0.01}$, as depicted in Fig. 3a. This uniform Ti-ZNSS layer encapsulates the ZNSS matrix. Further scrutiny, as shown in Figs. S6–S10, Supporting Information, revealed a profusion of nanoparticles uniformly dispersed around the ZNSS matrix. The excess of Ni in both the matrix and the Ti-ZNSS layer is a result of two factors: firstly, it is a consequence of our experimental design, and secondly, it reflects an inherent characteristic of ZrNiSn itself [45,46]. The EDS mapping images corroborate that these nanoparticles predominantly consist of $ZrO_2$. To cross-check these findings, we also conducted a thorough TEM-EDS analysis, focusing on the size and composition of these nanoparticles (Fig. 3b), which affirmed their $ZrO_2$ constitution.

From these observations, we infer the formation of multiscale composite phase structures, which encompasses $ZrO_2$ nanoparticles,

ranging from 50–100 nm in size, and the Ti-ZNSS layer, with a thickness of 50 to 500 nm (Fig. S8). These features are predominantly derived from the chemical interaction between the highly reactive amorphous $TiO_2$ layer and the ZNSS matrix during the SPS sintering process. Analogous phenomena have been observed in $Al_2O_3$-coated $ZrNiSn_{0.99}Sb_{0.01}$ and $TiO_2$-coated $Bi_2Te_{2.7}Se_{0.3}$ systems [25,44]. The underlying mechanism of this interface chemical reaction can be understood in terms of electronegativity and thermodynamic considerations. The electronegativity difference between Zr and O (2.11) is greater than that between Ti and O (1.9), hence favoring the bond formation between Zr and O over Ti and O. The formation energies of various oxides during high-temperature processes are critical determinants of their stability and presence in the final product. $ZrO_2$ exhibits a formation energy of −3.801 eV, which is lower than that of $TiO_2$ (−3.3 eV), $SnO_2$ (−2.123 eV), $NiO_2$ (−1.761 eV), and $Sb_2O_3$ (−1.728 eV). This indicates that $ZrO_2$ is thermodynamically the most favorable oxide under the experimental conditions. The detailed chemical reaction formula is delineated as follows:

$$ZrNi_{1.03}Sn_{0.99}Sb_{0.01} + xTiO_2 \xrightarrow{1373\,K} Zr_{1-x}Ti_xNi_{1.03}Sn_{0.99}Sb_{0.01} + xZrO_2$$

(1)

## Atomic scale characterization of the composite phase interface (ZNSS/ZrO₂)

The microstructural intricacies at the phase interfaces play a critical role in disentangling the electrical and thermal transport properties. To this end, a comprehensive microstructural examination of two key interfaces of the SPSed $TiO_2 = 3.2$ nm sample was conducted. Figure 4a–e illustrate the interface between the ZNSS matrix and $ZrO_2$ nanoparticles. Employing aberration-corrected STEM with atomic resolution, we discerned a pristine coherent interface between the $ZrO_2$ nanoparticles [100] and the ZNSS matrix [100]. The clarity of this interface is further validated by the inverse fast Fourier transform (IFFT) images, which exhibit a defect-free phase boundary. This immaculately coherent interface is crucial for sustaining a high carrier mobility $\mu_H$, thereby leading to excellent electrical properties. Furthermore, detailed examination of high-resolution $ZrO_2$ atomic images, alongside Fast Fourier Transform (FFT) data (Fig. S13), confirms the crystalline structure of $ZrO_2$ as monoclinic. This finding lays a solid theoretical foundation for investigating the interface potential barriers. A deeper analysis, integrating data from Fig. 4c and Fig. 4d, suggests that the impeccable coherence of the interface can be attributed to the closely matched lattice parameters of ZNSS (a = 3.105 Å) and the monoclinic $ZrO_2$ (a = 3.005 Å).

Combining geometric phase analysis (GPA) with ZNSS as the reference matrix, we were able to spatially capture the high $\varepsilon_{xy}$ strain within the $ZrO_2$ nanoparticles (Fig. 4e). Notably, the strain distribution in this context deviates from the norm observed in traditional nanocomposites, where strain typically accumulates around the phase boundary. Instead, in our sample, the strain is distributed throughout the entire particle. Further exploration through STEM HADDF (Fig. 4g) and ABF (Fig. 4h) phase images revealed abundant twin boundaries within the second-phase $ZrO_2$. Twinning within the $ZrO_2$ structure was confirmed by lattice splitting observed in the FFT analysis. Concurrently, GPA analysis demonstrated substantial strain in the $\varepsilon_{xx}$ direction at these twin boundary positions (Fig. 4i). This strain effectively impedes phonon propagation, resulting in a significant reduction in $\kappa_L$ [47]. Remarkably, the presence of two distinct coherent twin boundaries within the $ZrO_2$, each characterized by different rotation angles, as shown in Fig. 4j and k, underscores the sophisticated microstructural engineering. This intricate design is instrumental in achieving the decoupling of carrier and phonon transport.

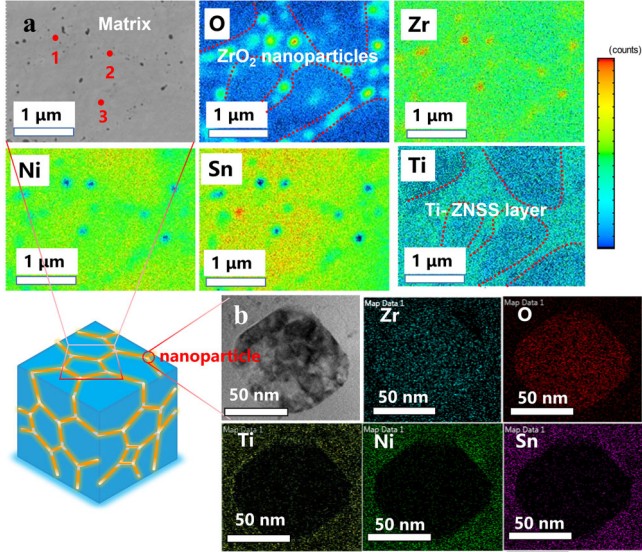

**Fig. 3 | Microscopic characterization of composite phase interfaces. a** EPMA backscattered electron image of $TiO_2 = 3.2$ nm sample and the corresponding EDS element of O, Zr, Ni, Sn, Ti maps (the color from blue to red indicates the content of elements from less to more). **b** Low magnification TEM image and corresponding EDS mapping images of the $ZrO_2$ nanoparticle.

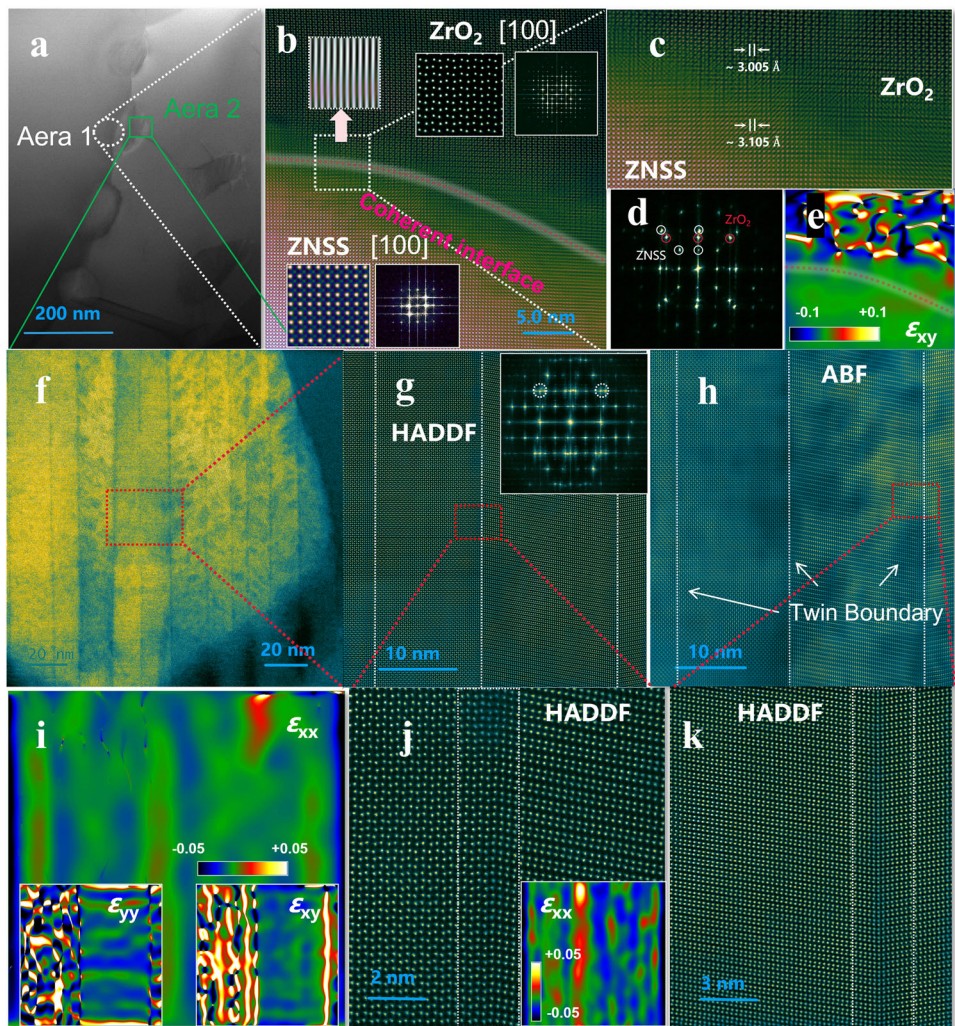

**Fig. 4 | Atomic scale characterization of the composite phase interface (ZNSS/ZrO₂).** The detailed features of the composite phase interface in the TiO$_2$ = 3.2 nm sample are as follows: A Low-magnification STEM-HADDF image (**a**) reveals ZrO$_2$ nanoparticles attached to grain boundaries. In the HADDF image of aera 1 (**b**) the atomic structure between ZrO$_2$ nanoparticles [100] and the ZNSS matrix [100] is distinctly visible. Further HADDF magnified images are presented in (**c**) with (**d**) and (**e**) showing their respective FFT image and GPA result. **f** STEM-HADDF image showing ZrO$_2$ precipitates. HADDF (**g**) and ABF (**h**) images respectively focus on the details of the ZrO$_2$ nanoparticles, with corresponding GPA results shown in **i**. Further HADDF magnified twin boundary images are presented in (**j**) and (**k**).

## Atomic scale characterization of the composite phase interface (ZNSS/Ti-ZNSS)

Figure 5a–d presents a detailed examination of the interface between the ZNSS matrix and the Ti-ZNSS layer. Elemental line scanning (Fig. 5a) indicates a heightened concentration of Ti around the grain boundaries, consistent with previous EDS results from EPMA. In regions with diminishing Ti concentration along the line scan (i.e., the interface between the Ti-ZNSS layer and the ZNSS matrix), a meticulous Z-contrast (Fig. 5b) discerns differences in mass between atoms, showcasing a flawlessly coherent interface established between the Ti-ZNSS layer and the ZNSS matrix, as confirmed by IFFT information[48]. As shown in Fig. 5c, d, advanced Z-contrast analysis of the HADDF images highlights pronounced darker features on the left side, conspicuously absent on the right. This phenomenon is attributed to the smaller atomic contrast exhibited by lighter Ti atoms compared to Zr. Consequently, through meticulous Z-contrast examination, a well-defined boundary delineates the Ti-ZNSS layer on the left from the ZNSS matrix on the right. To rigorously corroborate the accuracy of the data in Fig. 5d, a meticulous Z-contrast line scan is performed on the corresponding region in Fig. 5c. The observed trend reveals a sequential increase in atomic contrast from left to right, aligning

precisely with the contrast patterns depicted in Fig. 5d. This meticulous analysis serves as a stringent cross-verification, ensuring the reliability and consistency of the presented results.

## Electrical and thermal transport properties

In the context of the formation of coherent composite phase interfaces in TiO$_2$-coated ZNSS samples, a pertinent question emerges: Can these specialized phase interfaces facilitate the decoupling of carrier and phonon transport? Fig. 6 elucidates the electrical and thermal transport properties of these as-fabricated samples. A striking observation from Fig. 6a is the substantial enhancement of the Seebeck coefficient ($\alpha$) in ALD-coated ZNSS samples, compared to the initial ZNSS sample (Fig. 6a). In typical thermoelectric materials, an increase in the $\alpha$ is commonly accompanied by a decrease in the carrier concentration ($n_H$)[49]. However, as indicated by the Pisarenko curve of $\alpha$ - $n_H$ at room temperature (Fig. S16), the nearly constant $n_H$ before and after ALD coating indicates that the increase in $\alpha$ can only be attributed to the improvement in $m^*$ via the energy filtering effect[50,51]. Moreover, under the influence of coherent composite phase interfaces, the electrical conductivity ($\sigma$) of ALD-coated ZNSS samples does not show significant variation from the pristine uncoated sample (Fig. 6b). This

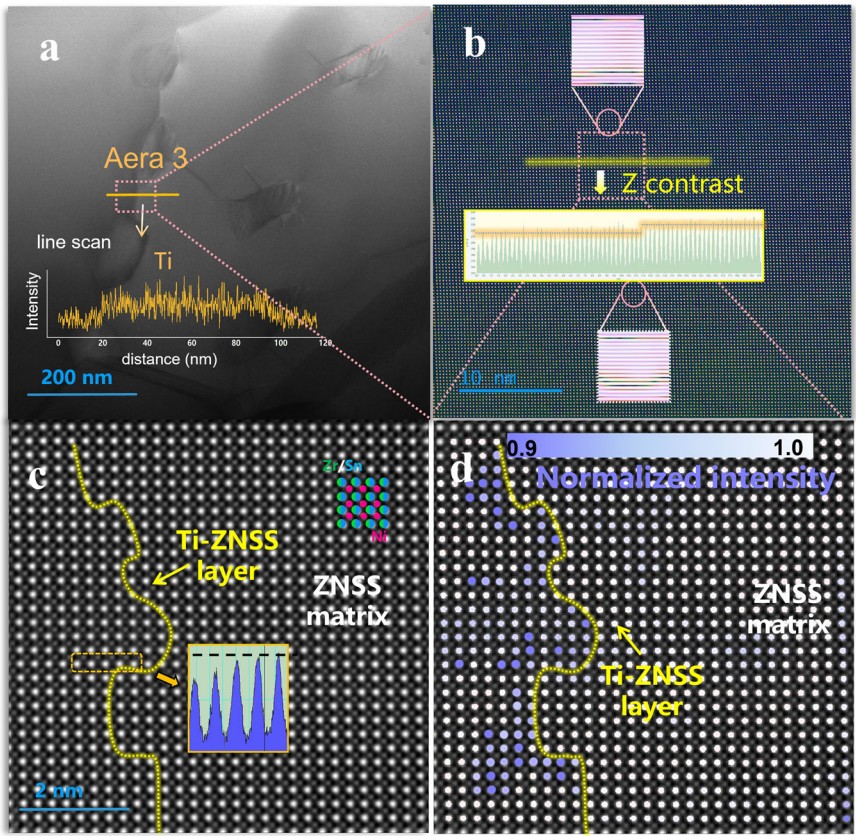

**Fig. 5 | Atomic scale characterization of the composite phase interface (ZNSS/Ti-ZNSS).** The yellow line in the (**a**) represents Ti element line scanning information, while (**b**) is an enlarged HADDF region within the pink box, and the inset displays the Z- contrast map of the yellow line. Further magnification of the HADDF atomic structure image is presented in (**c**) corresponding to the [100] HADDF image and normalized intensity of Ti columns in **d**.

stability in $\sigma$ can be predominantly attributed to the coherent phase interfaces, which are integral in maintaining $\mu_H$. In the case of $TiO_2 = 4.8$ nm and $TiO_2 = 6.4$ nm samples, the slight increase in the $\sigma$ is primarily due to the elevated $n_H$. Benefiting from the enhanced $\alpha$ and retained $\sigma$ via the coherent composite phase interfaces with multiple energy barrier, the power factor (*PF*) has been significantly enhanced after ALD coating (Fig. 6c). Taking the $TiO_2 = 3.2$ nm sample as an example, the simultaneously achieved high $\alpha$ of 204.1 $\mu VK^{-1}$ and $\sigma$ of 1448 $Scm^{-1}$ at 873 K give rise to a large *PF* of 60.24 $\mu Wcm^{-1}K^{-2}$. This attests to the efficacy of our phase interface engineering to improve the electrical properties of ZrNiSn-based materials.

Figure 6d depicts the temperature dependence of the thermal conductivity ($\kappa$) and lattice thermal conductivity ($\kappa_L$) for all the samples. A marked reduction in $\kappa$ is observed with increasing ALD coating layers. For instance, the room temperature $\kappa$ quickly drops from 5.41 $Wm^{-1}K^{-1}$ for C = 0 sample to 4.78 $Wm^{-1}K^{-1}$ for $TiO_2 = 6.4$ nm sample. Since the rational coherent phase interface design results in negligible changes in $\sigma$ across all the samples, this decline in $\kappa$ as increasing ALD coating layers can be largely ascribed to the diminished $\kappa_L$. Herein, $\kappa_e$ is evaluated by the Wiedemann-Franz law $\kappa_e = L\sigma T$ and $\kappa_L$ is obtained by subtracting the $\kappa_e$ from $\kappa$, where the Lorenz number $L$ is estimated via $L = 1.5 + exp(-|\alpha|/116)$[52]. As expected, the $\kappa_L$ substantially declines with increasing ALD coating layers over the entire temperature range (Fig. 6e). In particular, the lowest $\kappa_L$ of 1.72 $Wm^{-1}K^{-1}$ is attained in the $TiO_2 = 6.4$ nm sample, which is 20% lower than that of the pure ZNSS matrix. Ultimately, all ALD-treated samples exhibit superior *zT* values compared to the pristine ZNSS sample. Notably, the $TiO_2 = 3.2$ nm sample achieves an advanced peak *zT* value of approximately 1.3 (Fig. 6f).

## Analysis of electrical and thermal transport mechanisms

To elucidate the synergistic mechanism underlying the observed enhancement of thermoelectric performance, we employed the single parabolic band model (SPB) to generate Pisarenko curves for samples with different numbers of ALD cycles (Fig. 7a), with experimental data at 300 K and 800 K distinctly annotated. Initially, an analysis of the variation in $m^*$, as depicted in Fig. 7b, reveals a crucial insight. The intrinsic band structure of the ZNSS matrix remains highly stable across temperatures, with $m^*$ being 2.7 $m_e$ at both 300 K and 800 K. This finding aligns with the prior research conducted by Zhu et al.[53]. Furthermore, an intriguing trend is observed: regardless of at the temperature (300 K or 800 K), the $m^*$ demonstrates a gradual increase with an escalating number of ALD cycles. Most intriguingly, at 800 K, the $m^*$ for all ALD-treated samples is significantly elevated compared to their 300 K counterparts. According to the research conducted by Snyder et al.[50], the interface potential barriers between nanoinclusions and the matrix can selectively filter out low-energy charge carriers, thereby enhancing the DOS near the Fermi level (i.e., $m^*$) and consequently improving the $\alpha$. Moreover, Zhang et al. proposed that the enhancement in $m^*$ resulting from a single interface potential barrier tends to diminish at elevated temperatures[23]. The theoretically optimal estimation of $\Delta E \sim 1-10$ $k_BT$ supports notion that constructing hierarchical potential barriers could effectively mitigate this issue[31]. Therefore, the construction and optimization of potential energy barriers at the interfaces between the composite phases and the ZNSS matrix are critical to the enhancement of the electrical transport properties.

Given the composite phase structures prepared by ALD, it is imperative to further investigate the details of the two distinct types of

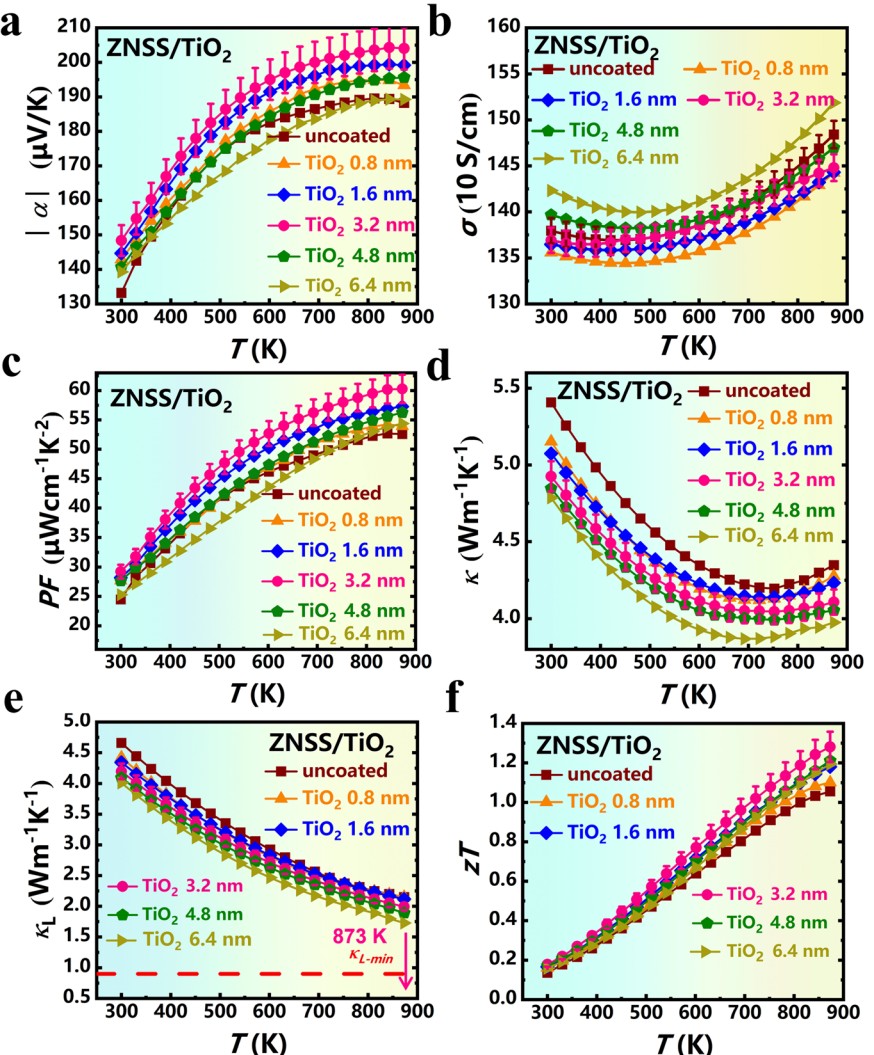

**Fig. 6 | Electrical and thermal transport properties.** The electrical and thermal properties of the ZNSS samples before and after different ALD cycles, where uncoated, $TiO_2$ = 0.8 nm, $TiO_2$ = 1.6 nm, $TiO_2$ = 3.2 nm, $TiO_2$ = 4.8 nm, $TiO_2$ = 6.4 nm.

**a** $\alpha$, **b** $\sigma$, **c** $PF$, **d** $\kappa$, **e** $\kappa_L$ and **f** $zT$. Error bars were estimated from the repeatability of the experimental result; three measurements were carried out for each material.

$\Delta E$ at the Ti-ZNSS/ZNSS and ZrO$_2$/ZNSS interfaces. After calibrating the work function for each band structure, an interface $\Delta E$ of approximately 0.2 eV was observed between the conduction band minimum (CBM) at the X-point between ZrNiSn, Zr$_{0.67}$Ti$_{0.33}$NiSn and TiNiSn (Fig. S13). This observation aligns with the the chemical reaction dynamics outlined in Equation 1, revealing that with an increasing number of ALD coating layers, there is a corresponding rise in Ti content in the Ti-ZNSS layer. Consequently, this leads to a tunable $\Delta E$ between the Ti-ZNSS layer and the ZNSS matrix, ranging approximately from 0 to 0.20 eV. In contrast, the band structure analysis of the monoclinic ZrO$_2$ indicated a higher $\Delta E$, approximately 0.6 eV (Fig. S14). This differential in $\Delta E$ values for the two interfaces, at approximately 0.2 eV and 0.6 eV respectively (Fig. 7c), fits well within the theoretically optimal range of $\Delta E \sim 1\text{-}10\ k_BT$ (0.25 eV/300 K – 0.75 eV/873 K)[31].

Furthermore, Fig. 7d distinctly demonstrates the remarkable elevation in the $m^*$ for the ALD-treated samples, observable at both ambient and elevated temperatures. The enhancement is more pronounced at higher temperatures. The underlying rationale for this phenomenon becomes apparent when juxtaposed with the semiconducting behavior of single $\Delta E$-induced Half-Heusler (HH)/Full Heusler (FH) samples[54]. Particularly at 800 K, the ratio of $m^*$ (with $\Delta E$) to $m^*$ (without $\Delta E$) displays a diminishing trend relative to 300 K, highlighting a limitation inherent in a singular energy barrier system.

Hence, the $PF$ (Fig. 7e) reveals a clear and consistent enhancement across all ALD-treated samples than that in pristine ZNSS sample, evident at both 300 K and 873 K. Such a phenomenon is a testament to the strategic construction of interfaces through ALD, offering a compelling insight into the intricate dynamics governing the thermoelectric enhancement observed in our experimental samples.

From the aspect of defect physics, the introduction of a second phase with a distinct interface from the matrix is conventionally perceived as a two-dimensional defect, imparting scattering interactions on both phonons and carriers[55,56]. However, coherent phase interfaces are recognized as a distinctive subclass of two-dimensional defects[57,58]. Notably, these interfaces are remarkable for their ability to scatter phonons while simultaneously preserving high $\mu_H$, rendering them a specialized and nuanced facet in the landscape of material defects. Therefore, the coherent composite phase interfaces constructed via ALD exhibit a comprehensive integration of this particular attribute. As depicted in Fig. 7f, a discernible pattern is observed with the continual increase in ALD cycles. At 300 K, $\mu_H$ remarkably consistent, while there is a notable successive reduction in $\kappa_L$. This observation underscores the meticulous orchestration of these interfaces in simultaneously upholding $\mu_H$ and modulating $\kappa_L$, highlighting the efficacy of ALD-driven fabrication of coherent composite phases.

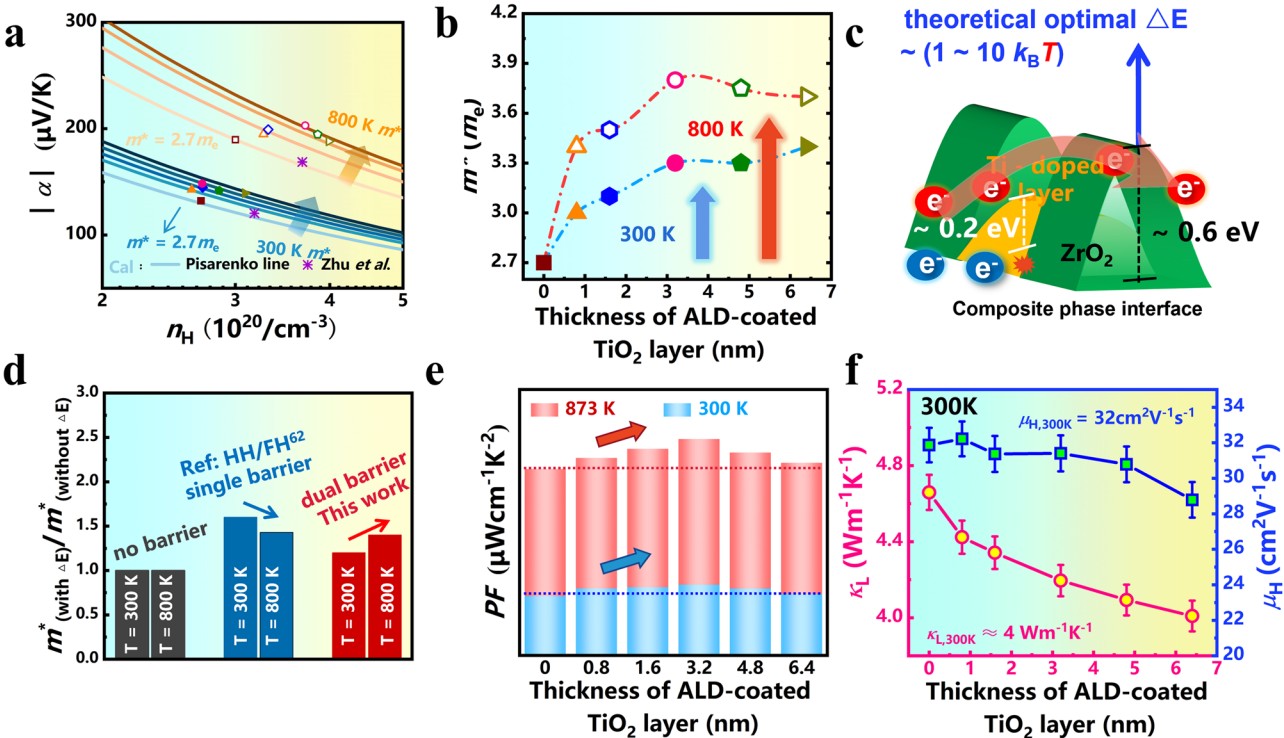

**Fig. 7 | Electrical and thermal transport mechanisms.** Temperature dependence of (**a**) Pisarenko relationship of $\alpha$ and component-dependent $m^*$. **b** at 300 and 800 K for the ZNSS samples before and after different ALD cycles, where uncoated, $TiO_2 = 0.8$ nm, $TiO_2 = 1.6$ nm, $TiO_2 = 3.2$ nm, $TiO_2 = 4.8$ nm, $TiO_2 = 6.4$ nm. **c** Schematic diagram of the energy filter effect of composite phase interface on electron transport. **d** The ratio of $m^*$ (with $\Delta E$) to $m^*$ (without $\Delta E$) at 300 K and 800 K for cases no barrier, with a single energy barrier, and with dual energy barriers[54]. **e** Component-dependent $PF$ at 300 and 873 K for the ZNSS samples before and after different ALD cycles, where uncoated, $TiO_2 = 0.8$ nm, $TiO_2 = 1.6$ nm, $TiO_2 = 3.2$ nm, $TiO_2 = 4.8$ nm, $TiO_2 = 6.4$ nm. **f** The $\mu_H$ and $\kappa_L$ at room temperature for $TiO_2$-coated ZNSS samples.

## Thermoelectric performance comparison

The remarkable advancements in elevating $m^*$, preserving $\mu_H$, and attenuating $\kappa_L$ through coherent composite phase interface engineering have unequivocally established this methodology as a cornerstone in advancing thermoelectric performance. Figure 8a, b vividly illustrate that the $PF$ achieved via the coherent composite interfaces significantly outperforms those observed in $n$-type ZrNiSn systems. Concurrently, the multidimensional defects meticulously constructed via composite phase interface engineering exert a profound constraint on phonon propagation, resulting in a marked reduction in $\kappa_L$. Consequently, this sample manifests a remarkable average $zT$ value of 0.73 within the temperature range of 300 to 873 K (Fig. 8c). The relevance of this finding is amplified when considering the critical benchmarks for assessing thermoelectric materials: the average $zT$ and average power factor. The average $zT$ dictates the device's conversion efficiency, whereas the average power factor governs the device's output power. In this context, as illustrated in Fig. 8d, the $TiO_2 = 3.2$ nm sample not only demonstrates a significant improvement, almost twofold, compared to the original ZrNiSn sample, but it also stands out remarkably against other high-performance n-type half-Heusler thermoelectric materials reported in contemporary research. These results demonstrate the effectiveness of coherent composite phase engineering in decoupling carrier and phonon transport in thermoelectric materials.

## Discussion

In the context of coherent composite phase interface engineering, a Ti-ZNSS layer (aka $Zr_{1-x}Ti_xNi_{1.03}Sn_{0.99}Sb_{0.01}$) with uniformly distributed $ZrO_2$ nanoparticles has been successfully constructed based on the ALD technique to decouple the interrelated thermoelectric parameters in $n$-type ZrNiSn. The composite phase structures including the Ti-ZNSS layer and $ZrO_2$ nanoparticles are achieved through an interfacial chemical reaction between the ZNSS matrix and amorphous $TiO_2$ during SPS process, where amorphous $TiO_2$ is deposited on the ZNSS substrate surface in advance via ALD technique adopting with TDMAT and $H_2O$ as precursors. The constructed composite interfaces encompass energy barriers localized at the Ti-ZNSS layer/ZNSS and $ZrO_2$/ZNSS interfaces. This strategic arrangement facilitates effective scattering of low-energy electrons across distinct temperature regimes, thereby elevating the density-of-states effective mass at both ambient and elevated temperatures. Furthermore, the imposition of the flawless coherent phase interfaces between Ti doped layer/ZNSS and $ZrO_2$/ZNSS adeptly scatter phonons at the composite phase interface, while concurrently preserving decent carrier mobility. Combining with the ultrahigh $PF \sim 60.24$ $\mu Wcm^{-1}K^{-2}$ and ultralow $\kappa_L \sim 1.72$ $Wm^{-1}K^{-1}$, a record high $zT \sim 1.3$ and average $zT_{avg}$ of $\sim 0.73$ between 300 and 873 K is attained in $TiO_2 = 3.2$ nm sample. These results not only enrich our comprehension of phase interface dynamics, but also significantly contribute to the emerging paradigm of "coherent composite phase interface engineering" in thermoelectric and functional materials. Another potential application worth our attention is: By employing ALD technology to precisely modify the interfaces, it is not only possible to prevent Ostwald ripening, but also promising to significantly enhance the stability of thermoelectric materials under long-term operational conditions.

## Methods

Polycrystalline samples $ZrNi_{1.03}Sn_{0.99}Sb_{0.01}$ matrix was prepared by the levitation melting-ballmilling-spark plasma sintering (SPS). Alloys were first prepared by levitation melting the stoichiometric amounts of Zr (piece, 99.99%), Ni (rods, 99.99%), Sn (block, 99,99%), and Sb (block, 99.99%) under an argon atmosphere for 2 min, and the melt was

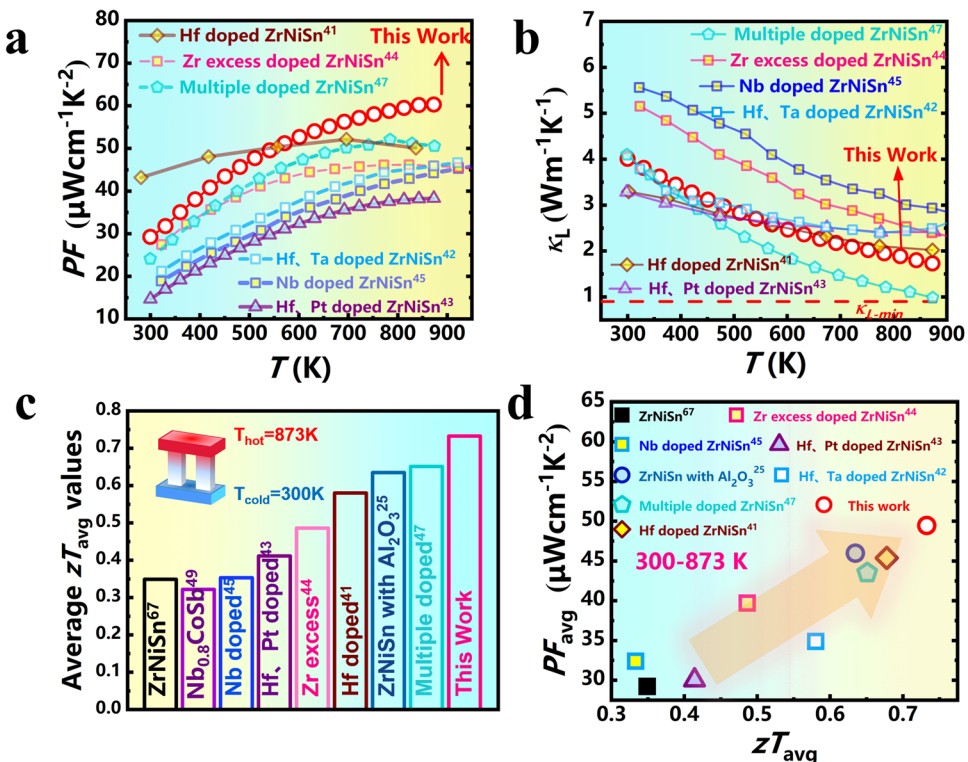

**Fig. 8 | Comparison of electrical and thermal transport performance.** Comparison $PF$ (**a**) and $\kappa_L$ (**b**) between $TiO_2$ = 3.2 nm sample and other advanced HH alloys[59–64]. **c** Comparison on $zT_{avg}$, **d** $zT_{avg}$ and $PF_{avg}$ between $TiO_2$ = 3.2 nm sample and other advanced HH alloys, where the hot-side $T_{hot}$ and cold-side $T_{cold}$ temperature is 873 K and 300 K, respectively[59–65,67].

quenched in a water-cooled copper crucible. The ingots were remelted six times to ensure homogeneity. The ballmilling process for 1 h at 800 rpm under an argon atmosphere. The fine powders were to use as the matrix of ALD.

A $TiO_2$ layer was coated on the surface of the $ZrNi_{1.03}Sn_{0.99}Sb_{0.01}$ matrix in a homemade continuous-flow ALD reactor operated under a base pressure of ~1 Torr. Typically, the 8 g as-prepared $ZrNi_{1.03}Sn_{0.99}Sb_{0.01}$ matrix was transferred into an ALD chamber equipped with a vertical stainless rotating sample chamber. The $TiO_2$ layer was deposited using tetrakis(dimethylamido)titanium (TDMAT) and $H_2O$ as ALD precursors at 423 K. The bubbler containing TDMA was heated to 323 K and the delivery line was heated to 343 K. The precursor dose and purge time were 30 seconds and 60 seconds, respectively. Ar gas served as both a carrier and a purging gas. Then, the coated fine powders were sintered by Spark Plasma Sintering (SPS) at 1373 K under 60 MPa in vacuum for 10 min. The as-sintered samples, of which the relative density was >98%, were cut for thermoelectric property measurement and characterization.

The phase structure of coated powders and sintered samples were measured by X-ray diffraction (XRD) on a Rigaku Smartlab 9kw (tube voltage: 45 kV, tube current: 200 mA) diffractometer using Cu K a radiation ($\lambda$ = 1.5406 Å) and X-rayphotoelectron spectra (XPS) were acquired using a Microlab 350 surface analysis system equipped with a monochromatized Al anode X-ray source, pass energy is 20.0 eV. The chemical composition was obtained during electron probe microanalysis (EPMA, JOEL, JXA-8100) using wavelength dispersive spectroscopy (WDS), acceleration voltage of 300 kV and test beam current: 50 nA for surface analysis, 20 nA for quantitative testing. The TEM specimens of the sintered bulk materials were meticulously prepared utilizing a state-of-the-art focused ion beam system (FIB; JIB 4601 F, JEOL) to ensure precise cross-sectioning. Subsequently, the surface morphology and detailed crystallographic orientation were elucidated through high-resolution field-emission transmission electron

microscopy (HR-TEM; JEM-ARM300F), operated at an acceleration voltage of 300 kV to achieve optimal imaging conditions.

The sintered specimen was carefully sectioned into two distinct shapes: a rectangular prism measuring 3 mm by 3 mm by 11 mm, and a square prism with dimensions of 9.8 mm by 9.8 mm by 2 mm. The rectangular prisms underwent meticulous polishing to facilitate the measurement of the Seebeck coefficient ($\alpha$) and electrical conductivity ($\sigma$) using the ULVAC ZEM-3 apparatus, within a temperature span of 300 to 873 K. Subsequently, the overall thermal conductivity ($\kappa$) was deduced through the formula $\kappa = DC_p\rho$, where $D$ denotes the thermal diffusivity, $C_p$ is the specific heat at constant pressure, and $\rho$ is the material density. The thermal diffusivity coefficient ($D$) was ascertained using a Netzsch LFA 467-HT laser flash analyzer, across the temperature gradient of 300 to 873 Kelvin. Concurrently, the specific heat capacity ($C_p$) was determined employing a differential scanning calorimeter (DSC 404 F3), with a controlled heating rate of 10 K per minute. Sapphire comparison method, calibration procedure: blank test, sapphire test and sample test, each DSC curve is subtracted from the blank curve. The carrier density of sintered samples were measured using a physical property measurement system (PPMS, Quantum Design).

Electronic structure calculations of ZrNiSn, TiNiSn and $ZrO_2$ were carried out using the density functional theory (DFT) with the projector-augmented-wave (PAW) method as implemented in the Vienna Ab initio simulation package (VASP). The computational framework for our study is based on the Generalized Gradient Approximation (GGA), specifically incorporating the exchange-correlation functional formulated by Perdew, Burke, and Ernzerhof (PBE). The structural optimization of the models was performed using the conjugate gradient (CG) method, which was iterated until the maximum forces acting on the Hellmann-Feynman theorem and the total energy reached a threshold of $10^{-4}$ eV, indicating a stable configuration. The simulation parameters were carefully chosen, with a plane-wave cutoff

energy set at 450 electron volts to ensure the accuracy of the calculations. For the integration over the Brillouin zone (BZ) of the primitive unit cell, we employed a Monkhorst–Pack grid with a dense k-point sampling of $11 \times 11 \times 11$, which provides a comprehensive representation of the reciprocal space.

## Data availability

All data generated or analyzed during this study are included in the published article and its Supporting Information. The data that support the findings of this study are available from the corresponding author (wuhaijunnavy@xjtu.edu.cn) upon reasonable request.

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

## Acknowledgements

The work is supported by the National Key R&D Program of China (2021YFB3201100), the National Natural Science Foundation of China (52071218, 52172128), the Shenzhen Science and Technology Inno-vation Commission (20200731215211001, 20200814110413001), the Guangdong Basic and Applied Basic Research Foundation (2022A1515012492). The authors also appreciate the Instrumental Analysis Center of Shenzhen University. The authors would like to thank the strong support from Instrumental Analysis Center of Xi'an Jiaotong University, with special thanks to Chenyu Liang for his sup-port in the XRD tests.

## Author contributions

The paper was prepared though the contribution of all authors. L.H., H.W. and F.L. designed the work. Y.Z., S.L., K.C., J.W. prepared the sample and measured the thermoelectric transport properties. Y.Z., G.P., Z.Z., C.M., and S.G. performed structural nanocomposite char-acterization. G.P. and Z.Z. performed simulation. L.H., H.W., Y.Y, T.L., C.Z., X.D. and J.S. planed and supervised the work. Y.Z. and H.W. wrote the paper. All the authors edited the paper.

## Competing interests

The authors declare no competing interests.
