## [Peer Review File · Nature Communications]

Phase Interface Engineering Enables State-of-the-Art Half-Heusler ThermoelectricsREVIEWER COMMENTS

Reviewer #1 (Remarks to the Author):

The manuscript submitted by Zhang et al. demonstrates a phase interface engineering approach to enhance the thermoelectric properties of ZrNiSn-based half-Heusler compounds. The experimental design is quite adjective, and ALD-coated TiO₂ as a source reacted with ZrNiSnSb and formed ZrNiSnSb-ZrTiNiSnSb-ZrO₂ nanocomposite. In this case, a unique interface ZrNiSnSb-ZrTiNiSnSb-ZrO₂ with a suitable interface potential barrier is built up to scatter low-energy electrons, enhancing the Seebeck coefficient. A huge effort has been devoted to characterizing the interface microstructures, which helps understand the structure-properties relation and supports the argument of the energy filtering effect. It offers a new technical route to design the interface microstructure. However, I have several concerns regarding technical aspects that need addressing before considering this manuscript for publication in Nature Communications. My comments are as follows:

- 1) Many references are cited incorrectly. The authors need to double-check all references and cite them correctly.
- 2) In the introduction part, page 4, lines 89-90, "these methods fall short in precisely constructing coherent phase interface with multiple delta E". Well, it is hard to judge this argument. At least, all those cited methods (ball milling, hydrothermal, self-precipitation, and melt-spinning) are not aimed at creating the coherent phase interface with multiple delta E. To prove the authors' statement, they can try to ball mill ZrNiSnSb-(Zr_{0.67}Ti_{0.33})SnSb-ZrO₂ (same chemical composition as C=40, which I calculated according to the EMPA results in the supporting information).
- 3) To me, it is more a modulation doping. If TiO₂ can coat all matrix particles, then what you get after the SPS is a core-shell structure. How can you make sure TiO₂ can coat all matrix particles? How long will it take to coat the 8 g matrix?
- 4) It is nice to see the authors show that the matrix is a Ni-rich phase. The Ni-rich nature has been confirmed by neutron scattering experiments.
- 5) If the authors measured DSC to get C_p, do they observe the Sn melting peak in the DSC curve? Although the XRD does not show visible impurity, the SEM&EDS show the Sn-rich area (Figure 3a). Also, Ni is not uniformly distributed in Figure 3a.
- 6) Ti EDS mapping is missing in Figure 3b.
- 7) Temperature-dependent kL is missing in Figure 6. Figures 6 d and 6 e are the same.
- 8) It is assumed that the density functional theory (DFT) calculations were conducted for ZrNiSn, TiNiSn, and ZrO₂ at 0 K. However, these data can not accurately represent the band structure at 300 K and 800 K.
- 9) Besides, if we take the calculated band structures of ZrNiSn and TiNiSn, we can only conclude that the delta E between them is around 0.2. How can you conclude that the delta E between Zr_{0.67}Ti_{0.33}NiSn_{0.99}Sb_{0.01} and ZrNiSn_{0.99}Sb_{0.01} is in the range of 0-0.2?
- 10) The conductivity data has a certain trend with increased c numbers. I guess the conductivity is a tradeoff of the Ti doping (increase the n) and interface scattering (decrease the mobility)
- 11) I think modulation doping induced by adding TiO₂ is the key to enhancing the power factor.
- 12) If the method can be applied to the high ZT half Heusler compounds, this work will definitely deserve to be published in Nature Communications.

Reviewer #2 (Remarks to the Author):

Minor Revision. The authors introduced the enhancement in the thermoelectric performance of ZrNiSn-based alloys using atomic layer deposition technology, which effectively decouple the interrelated thermoelectric parameters in ZrNiSn. I have carefully read the manuscript, and I found that this is an interesting work with promising results, and the structure and logic are good. I am happy to recommend this work to be published in this journal after some revisions, since there are still unclear points in the current manuscript. My comments listed below may help the authors further improve their work:

- 1) The diagram in Figure 1a should be redrawn, such that the second phase is tens to hundreds of nanometers in size and should not be represented by small balls. The thickness of the Ti-doped second phases coating the ZNSS matrix should not be less than the interplanar spacing.
- 2) It can be seen from the XPS spectra in Figure S3 and EDS results in S8 that there is a lot of amorphous TiO₂ on the ZNSS surfaces, but no obvious changes in the diffraction peaks caused by amorphous TiO₂ as shown in Figure S4, whether it is due to the deduction of the XRD pattern background or the fact that the content of amorphous TiO₂ is less than the detection limit of XRD. The author needs further clarification.
- 3) It is difficult to explain the interface reaction mechanism based on STEM and HAADF results in Figure 2. This is because the HAADF image is essentially an atomic column contrast image, and its contrast is affected by many factors. On page 8 of the manuscript, the author's conclusion, "the residual Ti atoms occupy vacant Zr sites, which can create a Ti-ZNSS layer...", cannot be judged by the HAADF results alone. The STEM-EDS results should be combined to explain the exact mechanism.
- 4) On page 10 of the manuscript, if only Zr position elements change, the energy spectrum of other elements should be uniform. However, the distribution of elements of Ni and Sb is not very uniform in Figure 3a, so is it appropriate to judge the reaction equation from the experimental results in EPMA? The authors can add some energy spectrum analysis to verify the accuracy of the reaction formula.
- 5) Figure 6d and Figure 6e are repeated, and Figure 6e should be the lattice thermal conductivity. In addition, the test temperature when the minimum lattice thermal conductivity is reached should be added.
- 6) The point-scanned data in EPMA image of Table S1 and S2 show that the content of Ti in C20 and C40 samples is different with the increased of thickness of the deposited atomic layer.
The authors need to add the point-scanned data of the other composition ZNSS/C (C=10, 60, 80), because I believe there is a certain difference with the nominal chemical composition.
- 7) The serial numbers of the references in Figure 8 do not correspond to each other. Please check them carefully.
- 8) Error estimates for the different measured quantities can be included in experimental part or figures for better comprehension.

9) Kindly check the formatting of references and image resolution in accordance to journal standards. Such as there are some superfluous words (Adv. Energy Mater. 37/2021) in reference 17, the numbers in the material composition in references 5 and 17 should be small corner marks, and the resolution in Figure S1 is obviously poor. These mistakes should not have happened, and the authors should confirm them carefully before resubmitting.

10) The XPS fit peaks in Figure S3 are rough, especially the peaks near the 462 eV, so it is recommended to fit them after removing the background of XPS spectra.

11) For a good scientific paper, supporting documents and manuscript are equally important. However, the figures in the supporting files are rough, such as no picture numbers and ruler in Figure S9 Figure S11, and Figure S13.

12) Some outlooks should be added before the conclusion such as the future direction for practical applications.

Reviewer #3 (Remarks to the Author):

Zhang and coworkers have explored thermoelectric ZrNiSn using a range of advanced techniques, including aberration-corrected high-resolution microscopy. The authors have utilized an interesting idea to engineer the interfaces by inclusions of TiO₂ layers through atomically controlled processes. They have shown that scattering phonons against such interfaces, while maintaining high carrier mobility and high effective mass (energy filtering), is a potent approach to enhance thermoelectric efficiency (ZT). Consequently, the ZT value reaches 1.3 at 900 K, which is an important step towards broader applications. Such a coherent interface design is novel and deserves attention of the thermoelectric community and may have impact on other functional materials. The manuscript is well written. Hence, this manuscript should be accepted for publication after addressing the points raised below.

1) Since thermoelectric materials are often exposed to harsh conditions (atmosphere, high temperatures) under cyclic loading, a coherent interface may also be relevant to counteract these effects. Is it conceivable that further oxidation can be hindered by inclusion of TiO₂? Please discuss these issues in the manuscript. This would give a larger perspective.

2) The introduction is too short regarding the physical (transport) properties of ZnNiSn and related open questions to be tackled in the manuscript. Please provide a broader review on ZrNiSn, especially alternative pathways to enhance its thermoelectric performance (interface engineering with Al₂O₃ (Key Engineering Materials Vol. 249 (2003) pp 79-82) and amorphous configurations (Computational Materials Science 230 (2023) 112530)). Another reference regarding Al₂O₃ has been mentioned in the results section, but the introduction leaves the impression that interface engineering has not been attempted. Furthermore, why is TiO₂ a better choice than Al₂O₃? Could an amorphous interface (e.g., crystalline ZrNiSn interfaced with amorphous ZrNiSn) have a similar performance? Furthermore, the addition of Sb to ZrNiSn and selection of TiO₂ should also be more motivated.

3) ZrO₂ nanoparticles are obviously important. Please also discuss their role in the abstract.

4) Why is there only oxidation of Zr when TiO₂ is in contact with ZrNiSn-Sb? Why are Sn, Ni, and Sb inert? For instance, Sn can easily form oxides. Electronegativity, as mentioned on page 10, is only a partial cause. What about thermodynamics? Eq. (1) is not really supported.

- 5) How does the ZT value obtained in this work correlate with other Heusler systems? Please discuss.
- 6) Designating TiO₂ layers as “C = 0, C = 10, C = 20, C = 40, C = 60 and C = 80” is misleading. If somebody wants to reproduce these results (perhaps by using another technique), more general descriptors are needed (e.g., thickness rather than cycles). For example, C = 40 means that the TiO₂ thickness is about 3 nm (page 8). Why is this complicated designation (method oriented) necessary?
- 7) It would be useful to designate the binding energy of both contributions in Fig. S3. Please also compare the binding energy with literature.
- 8) Does hydrogen have any role here? Having H₂O as a precursor should not be ignored. In many fields, H is instrumental. Please discuss.
- 9) Fig. S4 has never been mentioned in the manuscript. If it's redundant, please remove it. Please check other figures in the supplementary materials as well.
- 10) Is the compositional analysis as accurate as presented in Table S1? Are all digits statistically reasonable to supply? What is the error bar? Why is O omitted in Table S1? Similar goes for Table S2.
- 11) The authors argue about strain in the interfaces. How is strain affecting the transport properties?
- 12) Please provide a reference for the procedure pertaining to the Pisarenko curve.
- 13) There are some typos in the manuscript (e.g., “fromthe” on page 16 and “Mpa” on page 23 (it should be MPa)). Please proofread the text.
- 14) Band structure in Fig. S10 is given for ZrNiSn and TiNiSn as well as ZrO₂ in Fig. S12, rather than the actual system investigated in the current work. Drawing conclusions from a very idealized model system(s) can be ambiguous. Please elaborate.
- 15) The statement that the “base pressure of ~ 1 Torr” was achieved in ALD experiments doesn't seem to be correct. This is probably the working pressure (1 Torr as a base pressure may give rise to full oxidation). Please state the base pressure (pressure before depositions).
- 16) More details are needed for the characterization techniques (e.g., power setting for XRD, pass energy for XPS, accelerating voltage for microscopy, calibration procedure for DSC, etc.).

Referee #1:

General comment: *The manuscript submitted by Zhang et al. demonstrates a phase interface engineering approach to enhance the thermoelectric properties of ZrNiSn-based half-Heusler compounds. The experimental design is quite adjective, and ALD-coated TiO₂ as a source reacted with ZrNiSnSb and formed ZrNiSnSb-ZrTiNiSnSb-ZrO₂ nanocomposite. In this case, a unique interface ZrNiSnSb-ZrTiNiSnSb-ZrO₂ with a suitable interface potential barrier is built up to scatter low-energy electrons, enhancing the Seebeck coefficient. A huge effort has been devoted to characterizing the interface microstructures, which helps understand the structure-properties relation and supports the argument of the energy filtering effect. It offers a new technical route to design the interface microstructure. However, I have several concerns regarding technical aspects that need addressing before considering this manuscript for publication in Nature Communications.*

General response: We appreciate the reviewer's positive comments. Please find our point-by-point responses below.

Detailed comment 1: “[1] Many references are cited incorrectly. The authors need to double-check all references and cite them correctly. ”

Response: Thank you for bringing this to our attention. We apologize for any confusion caused and have rigorously reviewed all references to ensure their accuracy. Each citation has been corrected to comply with the journal's formatting requirements, and we have verified the accuracy of each reference to uphold the integrity of our manuscript.

Detailed comment 2: “[2] In the introduction part, page 4, lines 89-90, “these methods fall short in precisely constructing coherent phase interface with multiple delta E”. Well, it is hard to judge this argument. At least, all those cited methods (ball milling, hydrothermal, self-precipitation, and melt-spinning) are not aimed at creating the coherent phase interface with multiple delta E. To prove the authors' statement, they can try to ball mill ZrNiSnSb-

(Zr_{0.67}Ti_{0.33})SnSb-ZrO₂ (same chemical composition as C=40, which I calculated according to the EMPA results in the supporting information). ”

Response: Thank you for your insightful suggestion. Following your advice, we conducted ball milling of the mixture of ZrNi_{1.03}Sn_{0.99}Sb_{0.01} and ZrO₂ to investigate the formation of a coherent interface. Based on the microstructural analysis presented in Fig. R1, it is evident that the ZrNiSn matrix and the ZrO₂ second phase do not exhibit a coherent interface. Additionally, experiments such as those by X. Y. Huang et al. (*Solid State Communications* **130**, 181–185, (2004)) involving the incorporation of ZrO₂ into ZrNiSn via ball milling have consistently shown a decrease in carrier mobility (μ_H), failing to achieve the ideal decoupling of electron and phonon transport. These results suggest that the introduction of ZrO₂ through ball milling is insufficient to create the desired coherent interface necessary for optimized electron and phonon transport properties. The ball milling process may lack the precision required to control the interface chemistry and structure, which are critical for achieving the desired thermoelectric (TE) performance.

In contrast, atomic layer deposition (ALD) has proven more effective in achieving a coherent interface between the ZrNiSn matrix and the ZrO₂ layer, as shown in Fig. 4. The ALD process allows for precise control over the thickness and uniformity of the ZrO₂ layer, which is crucial for forming a coherent interface that enhances the thermoelectric performance of the material.

We appreciate the opportunity to discuss these alternative methods and clarify the advantages of our chosen approach. We believe that our study provides valuable insights into the importance of interface engineering in developing high-performance thermoelectric materials.

Fig. R1. **a**, **b** and **c** HRTEM image of ZrO_2 nanoparticles composited under the ball milling process and corresponding EDS energy spectra. **e** and **f** show the detail view of the interface contact between ZrO_2 and $\text{ZrNi}_{1.03}\text{Sn}_{0.99}\text{Sb}_{0.01}$ in **d**.

Detailed comment 3: “[3] To me, it is more a modulation doping. If TiO_2 can coat all matrix particles, then what you get after the SPS is a core-shell structure. How can you make sure TiO_2 can coat all matrix particles? How long will it take to coat the 8 g matrix?”

Response: Thank you for your valuable comments. In this work, we utilized ALD to achieve a relatively uniform, atomic-scale coating of TiO_2 on the powder matrix materials prior to the spark plasma sintering (SPS) process. ALD is especially effective for coating powders with complex morphologies due to its gas-phase reactions, which ensure uniform deposition across even the most irregular surfaces.

During the ALD process, precursors in vapor form are introduced sequentially into a reaction chamber under vacuum. This approach allows for the deposition of shell materials with atomic-scale precision on powder samples, regardless of their morphological complexities.

Ideally, sintering the as-prepared powder with the core-shell structure results in polycrystalline materials with continuous heterogeneous grain boundaries.

The time required for the ALD process is determined by the number of cycles needed to achieve the desired coating thickness. For the deposition of the TiO₂ layer, we used Tetrakisdimethylamido-titanium (TDMAT) and H₂O as ALD precursors at 150 °C. The bubbler containing TDMAT was heated to 40 °C, and the delivery line was maintained at 70 °C. The precursor dose and purge times were set at 30 seconds and 60 seconds, respectively. Typically, one ALD cycle takes approximately 90 seconds.

The use of ALD for phase interface engineering not only enhances the thermoelectric performance of various materials but also demonstrates significant potential for industrial scalability and precision. This method is compatible with commercialization efforts due to its controlled and repeatable nature. We appreciate your inquiries and hope this explanation clarifies the methodology and rigor employed in our research.

For a detailed delineation of the distinctions between this work and conventional modulation doping, please refer to our response to detailed comment 11.

Detailed comment 4: “[4] It is nice to see the authors show that the matrix is a Ni-rich phase. The Ni-rich nature has been confirmed by neutron scattering experiments.”

Response: We appreciate the reviewers' acknowledgment of our demonstration that the matrix is a Ni-rich phase, as confirmed. ZrNiSn, a half-Heusler compound, is particularly intriguing due to its distinctive electrical and thermal transport properties, as well as the notable discrepancies between theoretical predictions and experimental results.

Neutron scattering proves invaluable for elucidating the macroscopic atomic arrangement while also detecting microscopic local disorders and defects. Previous studies (*J. Mater. Chem. C* **3**, 10534-10542 (2015), *J. Appl. Phys.* 116,163514 (2014)) utilizing neutron scattering have underscored the essential feature of ZrNiSn: the excess of Ni atoms. This Ni excess introduces local chemical disorder, which subsequently affects the electronic structure and, consequently, the thermoelectric performance of the material.

The respective content has been updated in the revised manuscript, as seen below: **The excess of Ni in both the matrix and the Ti-ZNSS layer is a result of two factors: firstly, it is a**

consequence of our experimental design, and secondly, it reflects an inherent characteristic of ZrNiSn itself.

Detailed comment 5: “[5] If the authors measured DSC to get Cp, do they observe the Sn melting peak in the DSC curve? Although the XRD does not show visible impurity, the SEM&EDS show the Sn-rich area (Fig. 3a). Also, Ni is not uniformly distributed in Fig. 3a.”

Response: Thank you for your insightful comment. We employed differential scanning calorimetry (DSC) to determine the heat capacity (Cp) of our samples, using sapphire as the reference material. DSC is highly sensitive to detecting minute heat flow changes associated with thermal events, which is essential for characterizing both physical and chemical transformations in materials.

We carefully selected three samples, each weighing approximately 50 milligrams, to minimize any discrepancies that might arise from sample heterogeneity. In Figure S17b, none of the samples exhibit a tin melting peak (around 504 K). In contrast to the Scanning Electron Microscopy (SEM) and Energy-Dispersive X-ray Spectroscopy (EDS) analyses described in Figure 3a, where Sn-rich regions were observed, it is determined that these regions do not represent the presence of elemental tin but rather small areas enriched with Sn-containing compounds. This enrichment is a result of slightly over adding tin during the preparation of the $\text{ZrNi}_{1.03}\text{Sn}_{0.99}\text{Sb}_{0.01}$ matrix using the suspension melting method to prevent tin evaporation. Because these areas are sufficiently small, their content is below the detection threshold of X-ray diffraction (XRD), which is typically around 5%. This is the reason why they are not visible in the XRD spectrum.

Furthermore, In addition, as shown in Fig. 3a, the distribution of Ni is relatively more heterogeneous, which is mainly due to the over-sensitivity of Electron Probe Micro-Analysis (EPMA) to different local compositions. Combined with the data from our spot-scanning (Table S3), it can be seen that the Ni in the matrix is about 0.3% more than that in the Ti-ZNSS layer, and that the relative amount of Ni is probably due to the interfacial reaction, where the Ni content disappears completely in the corresponding ZrO_2 nanoparticles. It is important to highlight that the melting point of Ni (1728 K) far exceeds the maximum testing temperature range of our DSC (up to 873 K). Thus, no melting peak for Ni could be observed within our DSC test conditions.

Additionally, we have provided logarithmic mode XRD spectra (Fig. S4b) to offer a clearer depiction of the phases present, confirming the absence of distinct Sn or Ni peaks within the detection limits of this technique.

Fig. S17. b Temperature dependent specific heat capacity C_p for ZNSS.

Fig. S4. b XRD pattern (log mode) of $\text{TiO}_2 = 3.2$ nm sample after SPS process.

Detailed comment 6: “[6] Ti EDS mapping is missing in Fig. 3b.”

Response: We apologize for the oversight regarding the absence of Ti EDS mapping in Fig. 3b. We have now included the Ti EDS map in the revised Fig. 2d to provide a comprehensive view

of its distribution within the sample. Your attention to detail and valuable feedback are greatly appreciated.

Fig. 2 ALD Synthesis Technology Route. **a** Schematic diagram illustrating the synthesis of the TiO₂ layer through ALD coating. **b** HRTEM images of ZNSS powders coated with 40 cycles of TiO₂. **c** Schematic representation of the interfacial reaction between the ZNSS matrix and ALD coating. **d** ABF images and corresponding (Zr, O, Ti element) EDS mapping of the bulk ZNSS sample coated with 40 cycles of TiO₂ (TiO₂ = 3.2 nm).

Detailed comment 7: “[7] Temperature-dependent κ_L is missing in Fig. 6. Figs 6d and 6e are the same.”

Response: Thank you for your meticulous review and valuable suggestions. We apologize for the oversight in Fig. 6 regarding the missing temperature-dependent κ_L . We have added the data in

the revised version. This revision enhances the clarity of the temperature dependence and overall understanding of the thermal properties discussed.

Fig. 6 Electrical and thermal transport properties. The electrical and thermal properties of the ZNSS samples before and after different ALD cycles, where uncoated, $\text{TiO}_2 = 0.8$ nm, $\text{TiO}_2 = 1.6$ nm, $\text{TiO}_2 = 3.2$ nm, $\text{TiO}_2 = 4.8$ nm, $\text{TiO}_2 = 6.4$ nm. **a** α , **b** σ , **c** PF , **d** κ , **e** κ_L and **f** zT .

Detailed comment 8: “[8] It is assumed that the density functional theory (DFT) calculations were conducted for ZrNiSn, TiNiSn, and ZrO₂ at 0 K. However, these data cannot accurately represent the band structure at 300 K and 800 K.”

Response: Thank you for your insightful comment. It is true that band structures can evolve with temperature changes, which is critical for accurately linking DFT results to experimental data at elevated temperatures like 300 K and 800 K.

In thermoelectrics, DFT calculations at 0 K are standard due to computational limitations and typically provide a baseline approximation of the electrical transport properties. To address temperature effects on the band structure, we rely on variable-temperature X-ray diffraction (XRD) to observe any significant changes in lattice parameters, indicative of thermal expansion and its impact on electronic properties.

Our variable-temperature XRD results indicate that the lattice parameters for our samples remain relatively invariant between 300 K and 800 K. This stability is likely due to the robust covalent bonds in ZrNiSn, resulting in a very low thermal expansion coefficient. The monoclinic phase of ZrO₂ similarly exhibits minimal lattice changes with temperature, as supported by *Physical Review B*, **70**, 245116 (2004). Based on these findings, we justified the application of 0 K band structure calculations for our thermoelectric materials at 300 K and 800 K.

Furthermore, we computed the density-of-states effective mass for ZrNiSn at these temperatures and confirmed it remains consistent at $2.7 m_e$, corroborating the high stability of the band structures as observed in studies by Prof. Tiejun Zhu and his team (*Sci. Rep.* **4**, 6888 (2014)).

This consistency across temperatures reinforces the validity of using 0 K DFT calculations to estimate material behaviors at higher temperatures within the specific context of this work. We acknowledge the limitations of this approach and continue to explore more precise methods to incorporate temperature-dependent changes in future work.

Fig. S5. a Variable temperature XRD patterns of $\text{TiO}_2 = 3.2 \text{ nm}$ sample and **b** the corresponding lattice parameters

Detailed comment 9: “[9] Besides, if we take the calculated band structures of ZrNiSn and TiNiSn , we can only conclude that the ΔE between them is around 0.2. How can you conclude that the ΔE between $\text{Zr}_{0.67}\text{Ti}_{0.33}\text{NiSn}_{0.99}\text{Sb}_{0.01}$ and $\text{ZrNiSn}_{0.99}\text{Sb}_{0.01}$ is in the range of 0-0.2?”

Response: Thank you for your critical observations and recommendations regarding our calculations of the band structures for ZrNiSn and TiNiSn , and their implications for the alloys $\text{Zr}_{0.67}\text{Ti}_{0.33}\text{NiSn}_{0.99}\text{Sb}_{0.01}$ and $\text{ZrNiSn}_{0.99}\text{Sb}_{0.01}$. We appreciate the opportunity to clarify these points further.

Regarding the ΔE calculations, the energy difference between ZrNiSn and TiNiSn is approximately 0.2 eV, as noted. For the alloys $\text{Zr}_{0.67}\text{Ti}_{0.33}\text{NiSn}_{0.99}\text{Sb}_{0.01}$ and $\text{ZrNiSn}_{0.99}\text{Sb}_{0.01}$, we infer the ΔE in the range of 0 to 0.2 eV based on linear interpolation between the two end-members (pure phases). This inference is based on the assumption that the partial substitution of Ti for Zr introduces a proportional but limited perturbation to the band structure compared to the end-member phases.

In response to your query, we have revised our analysis and supplemented it with an updated band structure diagram for $\text{Zr}_{0.67}\text{Ti}_{0.33}\text{NiSn}$. This new diagram more accurately depicts the conduction band minimum, clearly showing that it lies between those of ZrNiSn and TiNiSn . The inclusion of Ti exhibits a subtle upward shift in the conduction band minimum (around 0.1 eV),

which is critical for understanding the energy barrier modifications when designing materials for enhanced thermoelectric performance.

The primary effect of Sb doping is to optimize carrier concentration rather than significantly altering the band structure itself. Sb doping tends to raise the Fermi level to near or within the conduction band, enhancing electrical conductivity without a substantial shift in band structures. This aspect has been corroborated by previous studies involving Sb-doped ZrNiSn, which have shown minimal changes in the band features, thus supporting our assumptions and calculations (*Sci Rep.* **4**, 6888 (2014)).

Fig. S11. Band structure diagram of **a** ZrNiSn, TiNiSn and **b** $\text{Zr}_{0.67}\text{Ti}_{0.33}\text{NiSn}$.

Detailed comment 10: “[10] The conductivity data has a certain trend with increased *c* numbers. I guess the conductivity is a tradeoff of the Ti doping (increase the *n*) and interface scattering (decrease the mobility).”

Response: We appreciate the reviewer’s keen observation and valuable insights. Indeed, after applying over 40 cycles of ALD, there is a slight increase in the carrier concentration. This enhancement maybe due to the incorporation of additional Ti atoms into the ZNSS matrix during the TiO_2 deposition process, which mildly boosts the carrier concentration.

Regarding the observed decrease in carrier mobility, while coherent interfaces theoretically reduce carrier scattering, an excessive number of such interfaces—particularly with increased

ALD cycles—can act as scattering centers. This exacerbates carrier scattering and negatively impacts carrier mobility.

Despite these observations, the overall effect on electrical conductivity remains modest, primarily due to the balance between the benefits of Ti doping and the drawbacks of interface scattering. We have included a comprehensive discussion of these trade-offs in Fig. S19 in the revised supporting information.

Detailed comment 11: “[11] I think modulation doping induced by adding TiO_2 is the key to enhancing the power factor.”

Response: We appreciate the reviewer's insightful commentary. Modulation doping effectively facilitates the migration of charge carriers from doped to undoped zones, thereby reducing ionized impurity scattering and enhancing carrier mobility. This mechanism significantly improves electrical conductivity, a key determinant in the enhancement of PF .

Nevertheless, our observations indicated that the $C = 40$ sample displayed slightly lower electrical conductivity compared to the pristine $C = 0$ sample. This reveals that the increase in PF in this work primarily results from a substantial increase in the Seebeck coefficient across all measured temperatures, coupled with sustained high carrier mobility. The notable enhancement of the Seebeck coefficient and the preservation of carrier mobility are attributed to the presence of coherent dual-interface energy barriers, which represent a deviation from traditional modulation doping effects.

Moreover, while modulation doping is conventionally utilized to boost carrier mobility, its application extends beyond this to include sophisticated interface engineering. This approach can significantly refine the quality of semiconductor heterostructure interfaces. With an understanding of the broader potential of this technique, we are committed to exploring the underlying physical mechanisms of modulation doping and its extensive impact on thermoelectric performance in our future research endeavors.

Detailed comment 12: “[12] If the method can be applied to the high ZT half Heusler compounds, this work will definitely deserve to be published in *Nature Communications*.”

Response: We appreciate the reviewers' high regard for our work. We fully concur with your assessment, and indeed, this vision has propelled our innovative application of ALD technology

within the field of thermoelectrics. Our chosen matrix material, $\text{ZrNi}_{1.03}\text{Sn}_{0.99}\text{Sb}_{0.01}$, currently achieves a peak zT of approximately 1.1—one of the highest values reported for n-type ZrNiSn half-Heusler materials. Through the implementation of coherent composite phase interface engineering via ALD, we have successfully elevated the zT to a record 1.3 within this category.

In addressing thermal transport, ALD facilitates the formation of a consistent three-dimensional interface network, which effectively scatters phonons at grain boundaries. Furthermore, an aspect not previously underscored in our discussion is the ability of ALD to inhibit Ostwald ripening. This is crucial as it prevents grain growth and maintains the thermoelectric performance over extended periods. From an electronic transport perspective, ALD allows for the meticulous adjustment of interface barriers, thus optimizing the PF .

These findings affirm that ALD is not merely effective, but also exhibits significant versatility and scalability. This suggests promising potential for its application across a broader spectrum of materials beyond half-Heusler compounds. Encouraged by these outcomes, we intend to further explore these possibilities in our subsequent research initiatives.

Referee #2:

General comment: *“Minor Revision. The authors introduced the enhancement in the thermoelectric performance of ZrNiSn-based alloys using atomic layer deposition technology, which effectively decouple the interrelated thermoelectric parameters in ZrNiSn. I have carefully read the manuscript, and I found that this is an interesting work with promising results, and the structure and logic are good. I am happy to recommend this work to be published in this journal after some revisions, since there are still unclear points in the current manuscript. My comments listed below may help the authors further improve their work.”*

General response: We appreciate the reviewer’s positive comments. Please find our point-by-point responses below.

Detailed comment 1: *“[1] The diagram in Fig. 1a should be redrawn, such that the second phase is tens to hundreds of nanometers in size and should not be represented by small balls. The thickness of the Ti-doped second phases coating the ZNSS matrix should not be less than the interplanar spacing.”*

Response: We sincerely appreciate your insightful suggestions concerning Fig. 1a. Following your recommendations, we have revised the schematic diagram to more accurately depict the size and morphology of the second phase, ensuring it is represented within the range of tens to hundreds of nanometers, consistent with dimensions observed experimentally. Furthermore, we have adjusted the depiction of the Ti-doped layers to ensure that their thicknesses align with scientific accuracy, notably ensuring these thicknesses do not fall below the interplanar spacing.

Fig. 1 Schematic diagram of the decoupled thermoelectric properties of the working action of the coherent composite-phase. **a** Schematic diagram of electron and phonon transport in the multi-scale phase interfaces. **b** Temperature-dependent zT for the $TiO_2 = 3.2$ nm sample in this work, compared with those of other high- zT HH thermoelectric materials^{25,39-47}. **c** Schematic diagram of energy filtering effect. **d** Schematic diagram of electron and phonon transport in the coherent phase interfaces.

Detailed comment 2: “[2] It can be seen from the XPS spectra in Fig. S3 and EDS results in S8 that there is a lot of amorphous TiO_2 on the ZNSS surfaces, but no obvious changes in the diffraction peaks caused by amorphous TiO_2 as shown in Fig. S4, whether it is due to the

deduction of the XRD pattern background or the fact that the content of amorphous TiO₂ is less than the detection limit of XRD. The author needs further clarification.”

Response: We greatly value the reviewer's constructive critique. In response to your observations concerning the XPS spectra in Fig. S3 and the EDS results in Fig. S8, we confirm the presence of an amorphous TiO₂ layer on the ZNSS surfaces. This observation is consistent with the absence of significant changes in the XRD diffraction peaks displayed in Fig. S4.

The lack of discernible diffraction peaks in the XRD patterns for the amorphous TiO₂ can be attributed primarily to its non-crystalline nature, which inherently does not produce the sharp Bragg peaks typical of crystalline materials. Additionally, the thickness of the amorphous TiO₂ layer is less than 10 nm, a dimension that falls below the detection threshold of conventional XRD, which accounts for the absence of any discernible amorphous humps in the XRD patterns (*Nanoscale* **3**, 1580-1588 (2020)).

To provide further clarity within our manuscript, we have expanded upon the discussion regarding the formation mechanism of the amorphous TiO₂ layer and its detection limitations with XRD technology. A comprehensive explanation has been included in the main text to elucidate how the amorphous nature of TiO₂ and its minimal thickness contribute to the absence of detectable XRD signals.

Detailed comment 3: “[3] *It is difficult to explain the interface reaction mechanism based on STEM and HAADF results in Fig. 2. This is because the HAADF image is essentially an atomic column contrast image, and its contrast is affected by many factors. On page 8 of the manuscript, the author's conclusion, “the residual Ti atoms occupy vacant Zr sites, which can create a Ti-ZNSS layer...”, cannot be judged by the HAADF results alone. The STEM-EDS results should be combined to explain the exact mechanism.”*

Response: We appreciate the reviewer's insightful comments and have made significant revisions to our manuscript accordingly. We have enhanced our analysis by integrating high-angle annular dark-field (HAADF) imaging with energy-dispersive X-ray spectroscopy (EDS) results to offer a more comprehensive explanation of the interface reaction mechanism.

In the revised Fig. 2d, we have replaced the HAADF image with detailed EDS elemental mappings, clearly depicting the distribution of Zr, O, and Ti. This approach provides more robust evidence of the interactions at the interface. The EDS mappings reveal that residual Ti atoms

indeed occupy vacant Zr sites, forming a Ti-ZNSS layer. This observation not only corroborates our initial conclusion but also strengthens the scientific validity of our findings by presenting direct and compelling evidence of the proposed mechanism.

Fig. 2 ALD Synthesis Technology Route. **a** Schematic diagram illustrating the synthesis of the TiO₂ layer through ALD coating. **b** HRTEM images of ZNSS powders coated with 40 cycles of TiO₂. **c** Schematic representation of the interfacial reaction between the ZNSS matrix and ALD coating. **d** ABF images and corresponding (Zr, O, Ti element) EDS mapping of the bulk ZNSS sample coated with 40 cycles of TiO₂ (TiO₂ = 3.2 nm).

Detailed comment 4: “[4] On page 10 of the manuscript, if only Zr position elements change, the energy spectrum of other elements should be uniform. However, the distribution of elements of Ni and Sb is not very uniform in Fig. 3a, so is it appropriate to judge the reaction equation from the experimental results in EPMA? The authors can add some energy spectrum analysis to verify the accuracy of the reaction formula.”

Response: Thank you for raising this valuable question. We have provided further clarification on the rationale behind the chemical equation and the preferential reaction of Zr with O when in contact with TiO₂ and ZrNi_{1.03}Sn_{0.99}Sb_{0.01} matrix, based on several factors:

1. Chemical affinity and reactivity: Zr exhibits a stronger chemical affinity for O compared to Ti, Sn, Ni, and Sb. This chemical affinity is largely dictated by the electronegativity differences between the metal atoms and oxygen. Specifically, Zr and O have an electronegativity difference of 2.11, which is higher than those of Ti-O (1.90), Sn-O (1.48), Ni-O (1.53), and Sb-O (1.34). This greater difference enhances the likelihood of Zr bonding with O to form more stable compounds.

2. Thermodynamic considerations: The formation energies of various oxides during high-temperature processes are critical determinants of their stability and presence in the final product. ZrO₂ exhibits a formation energy of -3.801 eV, which is lower than that of TiO₂ (-3.3 eV), SnO₂ (-2.123 eV), NiO₂ (-1.761 eV), and Sb₂O₃ (-1.728 eV). This indicates that ZrO₂ is thermodynamically the most favorable oxide under the experimental conditions, explaining the preferential formation of this phase.

3. In-situ chemical dynamics: The ALD process promotes specific in-situ chemical reactions, particularly between the TiO₂ coating and the ZrNiSn matrix. This interaction facilitates the formation of ZrO₂ due to the reactivity of Zr with O, resulting in the creation of stable ZrO₂ nanoparticles. This dynamic is a key factor in the interface chemistry of our materials.

Regarding the non-uniform distribution of Ni and Sn observed in Fig. 3a, this can be attributed to the intentional design of our matrix. The matrix is Ni-rich, a strategy based on prior research that suggests stability benefits from an excess of Ni, as evidenced by neutron scattering experiments (*J. Mater. Chem. C* **3**, 10534-10542 (2015)). Similarly, the enrichment of Sn is due to a small excess of Sn present in the samples, this is a result of slightly overadding Sn during the preparation of the ZrNi_{1.03}Sn_{0.99}Sb_{0.01} matrix using the suspension melting method, in order to prevent the volatilization of Sn.

Detailed comment 5: “[5] Fig. 6d and Fig. 6e are repeated, and Fig. 6e should be the lattice thermal conductivity. In addition, the test temperature when the minimum lattice thermal conductivity is reached should be added.”

Response: We sincerely appreciate the reviewer's meticulous and insightful feedback. We apologize for the confusion caused by the duplication in Fig. 6d and 6e and the omission of specific details regarding the temperature-dependent lattice thermal conductivity (κ_L).

Upon reviewing the noted Fig. 6d and Fig. 6e, we have corrected Fig. 6e to properly display the κ_L as a function of temperature in the revised manuscript. The revised Fig. 6e now includes annotations that clearly indicate the temperature at which κ_L reaches its minimum value. These modifications have greatly enhanced the clarity and precision of our data presentation.

Fig. 6 Electrical and thermal transport properties. The electrical and thermal properties of the ZNSS samples before and after different ALD cycles, where uncoated, $\text{TiO}_2 = 0.8$ nm, $\text{TiO}_2 = 1.6$ nm, $\text{TiO}_2 = 3.2$ nm, $\text{TiO}_2 = 4.8$ nm, $\text{TiO}_2 = 6.4$ nm. **a** α , **b** σ , **c** PF , **d** κ , **e** κ_L and **f** zT . (The test temperature at which the minimum κ_L is observed is 873 K)

Detailed comment 6: “[6] The point-scanned data in EPMA image of Table S1 and S2 show that the content of Ti in C20 and C40 samples is different with the increased of thickness of the deposited atomic layer.

The authors need to add the point-scanned data of the other composition ZNSS/C (C=10, 60, 80), because I believe there is a certain difference with the nominal chemical composition.”

Response: We thank the reviewer for their insightful observations regarding the variation in Ti content across our samples. In response to your constructive suggestion, we have expanded our dataset to include point-scanned data for ZNSS/C samples with C = 10 and C = 60. These additional data are now comprehensively presented in the revised Supplementary Information (Tables S1 - S4).

The EPMA point scanning analysis at points 1 and 2 indicates that the Ti content at the Ti-ZNSS interface increases progressively with the number of ALD cycles. This trend substantiates the effective integration of Ti into the ZNSS matrix. At point 3, the matrix composition remains unaffected by the interface, displaying the original composition ($\text{ZrNi}_{1.03}\text{Sn}_{0.99}\text{Sb}_{0.01}$) without Ti. This consistency across multiple scans underscores the statistical robustness of our data.

Table S1. Point-scanned data in EPMA image of TiO₂ = 0.8 nm sample (1, 2 and 3 points).

position	Zr at%	Ni at%	Sn at%	Ti at%	Sb at%
1	30.375	33.755	32.895	2.602	0.373
2	30.672	33.304	32.693	2.969	0.362
3	32.157	34.237	33.221	0.000	0.381

Table S2. Point-scanned data in EPMA image of $\text{TiO}_2 = 1.6 \text{ nm}$ sample (1, 2, 3 points).

position	Zr at%	Ni at%	Sn at%	Ti at%	Sb at%
1	26.643	33.741	33.104	6.202	0.310
2	26.150	33.955	33.308	6.269	0.318
3	32.157	34.237	33.221	0.000	0.385

Table S3. Point-scanned data in EPMA image of $\text{TiO}_2 = 3.2 \text{ nm}$ sample (1, 2, 3 points).

position	Zr at%	Ni at%	Sn at%	Ti at%	Sb at%
1	23.770	33.141	32.928	9.874	0.287
2	23.526	33.315	33.897	9.969	0.293
3	33.154	33.554	32.971	0.000	0.321

Table S4. Point-scanned data in EPMA image of $\text{TiO}_2 = 4.8 \text{ nm}$ sample (1, 2, 3 points).

position	Zr at%	Ni at%	Sn at%	Ti at%	Sb at%
1	22.577	33.774	32.111	11.289	0.249
2	22.103	33.367	32.617	11.662	0.251
3	32.654	34.084	32.986	0.000	0.276

Detailed comment 7: “[7] The serial numbers of the references in Fig. 8 do not correspond to each other. Please check them carefully.”

Response: We sincerely apologize for the oversight regarding the mismatch of reference serial numbers in Fig. 8 and express our gratitude to the reviewer for identifying this issue. We have thoroughly reviewed and corrected the serial numbers of the references in the Fig. to ensure they are accurately aligned with the corresponding citations throughout the revised manuscript.

Detailed comment 8: “[8] Error estimates for the different measured quantities can be included in experimental part or Fig.s for better comprehension.”

Response: Good advice. After thorough consideration, we have made the following enhancements to the revised manuscript to improve the clarity and completeness of the data presentation:

In the main text, we have focused on presenting data for the optimally prepared sample ZNSS ($C = 40$). For this sample, we have now incorporated error bars into Fig. 6 to clearly delineate the uncertainty and variability in our measurements.

To ensure the clarity of all Fig.s while providing comprehensive error data, we have included detailed experimental data lines with error estimates for all other samples in Fig. S19. This allows us to present a full set of error estimates without overcrowding the main Fig.s, thereby improving the readability and accessibility of our data.

Fig. 6 Electrical and thermal transport properties with error lines. The electrical and thermal properties of the ZNSS samples before and after different ALD cycles, where uncoated, TiO₂ = 0.8 nm, TiO₂ = 1.6 nm, TiO₂ = 3.2 nm, TiO₂ = 4.8 nm, TiO₂ = 6.4 nm. **a** α , **b** σ , **c** PF , **d** κ , **e** κ_L and **f** zT .

Fig. S19 Electrical and thermal transport properties with error lines. The electrical and thermal properties of the ZNSS samples before and after different ALD cycles, where uncoated, TiO₂ = 0.8 nm, TiO₂ = 1.6 nm, TiO₂ = 3.2 nm, TiO₂ = 4.8 nm, TiO₂ = 6.4 nm. **a** α , **b** σ , **c** PF , **d** κ , **e** κ_L and **f** zT .

Detailed comment 9: “[9] Kindly check the formatting of references and image resolution in accordance to journal standards. Such as there are some superfluous words (*Adv. Energy Mater.* 37/2021) in reference 17, the numbers in the material composition in references 5 and 17 should be small corner marks, and the resolution in Fig. S1 is obviously poor. These mistakes should not have happened, and the authors should confirm them carefully before resubmitting.”

Response: We sincerely apologize for the oversight and have taken comprehensive measures to address these issues to enhance the quality of our submission:

Reference Formatting: We have thoroughly reviewed and corrected all references to ensure strict adherence to the journal's formatting requirements. Specifically, we have removed the superfluous words in reference 17 and adjusted the material composition numbers in references 5 and 17 to small corner marks as required.

Image Resolution and Fig. Quality: We acknowledge the poor resolution of Fig. S1 and have replaced it with a high-resolution version that meets the journal's standards. Additionally, we have carefully examined all other Figs in the revised manuscript and supplementary information to ensure they exhibit optimal clarity and consistency. Each Fig. has been meticulously revised to include clear scales, comprehensive descriptive text, and to maintain uniformity in Fig. numbering and layout.

Fig. S1. XRD pattern of the as-prepared ZrNi_{1.03}Sn_{0.99}Sb_{0.01} (ZNSS) powders.

Detailed comment 10: “[10] The XPS fit peaks in Fig. S3 are rough, especially the peaks near the 462 eV, so it is recommended to fit them after removing the background of XPS spectra.”

Response: Thank you for your valuable feedback. In response, we have addressed the concerns by removing the background noise and refining the peak fitting process, particularly around 462 eV. This adjustment has substantially improved the clarity and accuracy of our spectral analysis, leading to a more precise interpretation of the XPS data.

Fig. S3. XPS spectra of Ti 2p_{1/2} and Ti 2p_{1/3} core level for TiO₂ = 3.2 nm sample.

Detailed comment 11: “[11] For a good scientific paper, supporting documents and manuscript are equally important. However, the Fig.s in the supporting files are rough, such as no picture numbers and ruler in Fig. S9 Fig. S11, and Fig. S13.”

Response: We appreciate the reviewer's attention to detail and valuable feedback. In response, we have enhanced the quality of the Fig.s in our revised supplementary materials. Specifically, we have updated Fig.s S9, S11, and S13 by adding Fig. numbers and scales, ensuring clarity and consistency. These improvements significantly increase the readability and interpretability of the Fig.s, aligning with the high standards expected for scientific documentation.

Fig. S10. a HADDF, **b** ABF and **c** FFT images of matrix ZNSS.

Fig. S11. a HADDF and **b** ABF images of ZrO₂.

Fig. S14. a HADDF, b ABF and c1, c2, c3, c4, c5 GPA images of ZrO₂.

Detailed comment 12: “[12] Some outlooks should be added before the conclusion such as the future direction for practical applications.”

Response: We appreciate the reviewer's suggestion regarding the inclusion of future research directions. In response, we have expanded our discussion to underscore the potential practical applications of the ZrNiSn-based thermoelectric materials explored in this work.

ALD enables the creation of a uniform three-dimensional interface network, which significantly scatters phonons at interfaces and allows for the modulation of interface barriers. An important feature, previously not emphasized, is the capability of coated interface layers to prevent Ostwald ripening. This property is critical as it inhibits grain growth, thereby preserving the thermoelectric performance over extended durations. Given their robust thermoelectric properties and superior oxidation resistance, these materials are highly suitable for demanding

environments encountered in applications such as high-temperature waste heat recovery and power generation. Their suitability for such extreme conditions is bolstered by their consistent performance under high temperature and oxidative stress.

Referee #3:

General comment: “Zhang and coworkers have explored thermoelectric ZrNiSn using a range of advanced techniques, including aberration-corrected high-resolution microscopy. The authors have utilized an interesting idea to engineer the interfaces by inclusions of TiO₂ layers through atomically controlled processes. They have shown that scattering phonons against such interfaces, while maintaining high carrier mobility and high effective mass (energy filtering), is a potent approach to enhance thermoelectric efficiency (ZT). Consequently, the ZT value reaches 1.3 at 900 K, which is an important step towards broader applications. Such a coherent interface design is novel and deserves attention of the thermoelectric community and may have impact on other functional materials. The manuscript is well written. Hence, this manuscript should be accepted for publication after addressing the points raised below.”

General response: We appreciate the reviewer’s positive comments. Please find our point-by-point responses below.

Detailed comment 1: “[1] Since thermoelectric materials are often exposed to harsh conditions (atmosphere, high temperatures) under cyclic loading, a coherent interface may also be relevant to counteract these effects. Is it conceivable that further oxidation can be hindered by inclusion of TiO₂? Please discuss these issues in the manuscript. This would give a larger perspective.”

Response: Thank you for your insightful comments. In this work, we utilized atomic layer deposition (ALD) to apply a uniform TiO₂ coating on the ZrNiSn matrix, which significantly enhanced the oxidation resistance of the ZrNiSn powders, as illustrated in Fig. 2b.

During the high-temperature sintering process, this TiO₂ coating reacted in situ with the ZrNiSn matrix, forming a Ti-ZNSS layer and ZrO₂ nanoparticles. The Ti-ZNSS layer and ZrO₂, renowned for its stability, further bolsters the material's resistance to oxidation. Consequently, samples with the TiO₂ coating exhibited superior oxidation resistance at elevated temperatures compared to the uncoated ZrNiSn matrix.

Following your suggestion, we have expanded our discussion in the revised manuscript to further explore the potential of the TiO₂ layer to enhance the material’s durability and inhibit further oxidation, particularly under severe conditions and cyclic loading. This enhancement broadens the scope of these thermoelectric (TE) materials' applications in challenging environments.

Detailed comment 2: “[2] The introduction is too short regarding the physical (transport) properties of ZrNiSn and related open questions to be tackled in the manuscript. Please provide a broader review on ZrNiSn, especially alternative pathways to enhance its thermoelectric performance (interface engineering with Al₂O₃ (*Key Engineering Materials* Vol. 249 (2003) pp 79-82) and amorphous configurations (*Computational Materials Science* 230 (2023) 112530)). Another reference regarding Al₂O₃ has been mentioned in the results section, but the introduction leaves the impression that interface engineering has not been attempted. Furthermore, why is TiO₂ a better choice than Al₂O₃? Could an amorphous interface (e.g., crystalline ZrNiSn interfaced with amorphous ZrNiSn) have a similar performance? Furthermore, the addition of Sb to ZrNiSn and selection of TiO₂ should also be more motivated.”

Response: Thanks. The reviewer raised several profound questions. We would like to answer them together as follows.

ZrNiSn, an n-type thermoelectric material, exhibits a substantial density-of-states effective mass ($m^* \sim 2.7 m_e$) and a low deformation potential. These characteristics enable it to maintain a high Seebeck coefficient while retaining decent carrier mobility (μ_H), resulting in a high power factor. However, the high lattice thermal conductivity (κ_L) limits its thermoelectric performance. To mitigate this, alloying has proven effective in enhancing the zT by reducing κ_L . Additionally, nanostructuring strategy, such as grain refinement and increased grain boundary scattering, have been employed to scatter phonons more effectively. Interface engineering, including incorporating Al₂O₃ through ball milling, has successfully reduced κ_L and improved zT (*Key Engineering Materials* **249**, 79-82 (2003)). Furthermore, introducing amorphous ZrNiSn has achieved ultralow κ_L (*Computational Materials Science* **230**, 112530 (2023)). Despite these advancements, these approach did not enhance the electrical PF due to decreased μ_H and, consequently, electrical conductivity. Thus, the coupled nature of electron and phonon transport properties limits the extent of zT improvement.

In this work, we adopted a composite coherent interface engineering strategy aimed at effectively scattering phonons while minimizing the reduction in μ_H . By leveraging the energy filtering effect, we aimed to enhance the density-of-states effective mass and achieve synergistic optimization of electron and phonon transport properties, significantly improving the zT of ZrNiSn-based thermoelectric materials.

Regarding the choice of Al₂O₃ and TiO₂ as coating materials, Al₂O₃ was initially considered due to its high melting point and chemical stability. However, high-temperature phase interface reactions revealed that the Al element's stability is inferior to that of the Ti element under similar conditions. XPS analysis indicated that after sintering at 1100 °C, the Al₂O₃ coating often exhibited Al element deficiency, likely due to Al volatilization or complex reactions with the matrix (*J. Mater. Chem. A* **7**, 26053–26061 (2019)). Conversely, TiO₂ demonstrated better stability during high-temperature sintering. The Ti element is less volatile and more likely to form a stable solid solution with the ZrNiSn matrix. We observed Ti doping into the interface layer, enhancing interface stability and optimizing the band structure, thereby improving thermoelectric performance.

Regarding Sb doping, our primary goal was to optimize the carrier concentration (n_H) of the ZrNiSn matrix. As a donor dopant, Sb effectively increases the n_H from approximately 10¹⁹ cm⁻³ to 10²⁰ cm⁻³, which is crucial for improving the PF . Building on Sb doping, we further facilitated the co-optimization of both carrier and phonon transport properties, substantially enhancing the zT through composite coherent interface engineering. This strategy ensures a balanced improvement in thermoelectric performance by effectively managing the interplay between electrical and thermal transport.

These discussions have been incorporated into the revised introduction Page 3 - 4 (70-73 lines).

The respective content has been updated in the revised manuscript, as seen below: **For example, the introduction of Al₂O₃ into ZrNiSn or the creation of amorphous ZrNiSn have both successfully led to a significant reduction in κ_L . However, these approaches did not effectively enhance PF due to the decrease in μ_{hi} . Consequently, the coupling between electronic and phonon transport properties has limited the potential for improving the figure of merit zT .**

Detailed comment 3: “[3] ZrO₂ nanoparticles are obviously important. Please also discuss their role in the abstract.”

Response: We appreciate the reviewer's insight regarding the critical role of ZrO₂ nanoparticles. ZrO₂ is instrumental, particularly for its effects on phonon scattering. Our observations confirm

the presence of numerous twin boundaries within the ZrO₂, which significantly enhances phonon scattering and thereby effectively reduces κ_L .

Moreover, we have engineered a coherent interface between the ZrO₂ nanoparticles and the ZnSS matrix via an in-situ chemical reaction. The establishment of coherent interfaces not only further diminishes the κ_L but also mitigates potential negative impacts on carrier transport.

Acknowledging the constraints on word count in abstracts, we have succinctly outlined the role of ZrO₂ in the revised abstract to underscore its significance. A more detailed discussion on the implications of ZrO₂ for enhancing thermoelectric performance is comprehensively presented in the introduction and related work sections of the manuscript Page 5 (117-119 lines).

The respective content has been updated in the revised manuscript, as seen below: **Furthermore, we are pleasantly surprised to discover that there are a significant number of twins within the ZrO₂ nanoparticles. This twin structure provides an additional scattering center for phonon transport at the nanoscale.**

Detailed comment 4: “[4] Why is there only oxidation of Zr when TiO₂ is in contact with ZrNiSn-Sb? Why are Sn, Ni, and Sb inert? For instance, Sn can easily form oxides. Electronegativity, as mentioned on page 10, is only a partial cause. What about thermodynamics? Eq. (1) is not really supported.”

Response: We thank the reviewer for this insightful comment. The preferential oxidation of Zr, despite the presence of other metals like Ti, Sn, Ni, and Sb, can be attributed to a combination of factors influencing the chemical reactivity and thermodynamic stability during the synthesis process:

1. Chemical affinity: Zr exhibits a stronger chemical affinity for O compared to Ti, Sn, Ni, and Sb. This affinity is largely dictated by the electronegativity differences between the metal atoms and oxygen. Specifically, Zr and O have an electronegativity difference of 2.11, which is higher than those of Ti-O (1.90), Sn-O (1.48), Ni-O (1.53), and Sb-O (1.34). This greater difference enhances the likelihood of Zr bonding with O to form more stable compounds.

2. Thermodynamic stability: From a thermodynamic perspective, ZrO₂ is more stable than the oxides of the other metals involved. During the high-temperature sintering process, the formation energies of the oxides are crucial in determining their formation likelihood. ZrO₂ has a formation energy of -3.801 eV, which is notably lower than those of TiO₂ (-3.3 eV), SnO₂ (-

2.123 eV), NiO₂ (-1.761 eV), and Sb₂O₃ (-1.728 eV). The lower formation energy of ZrO₂ indicates a higher thermodynamic propensity for its formation.

3. Interface energy considerations: The synthesis process fosters a coherent interface between ZrO₂ and the ZrNiSn matrix. This interface, being energetically favorable, further facilitates the selective formation of ZrO₂ nanoparticles at the interface due to reduced interface energy.

These factors collectively explain why Zr reacts preferentially with O during the synthesis, leading to the predominant formation of ZrO₂, while the other elements remain relatively inert under the same conditions.

Detailed comment 5: “[5] How does the zT value obtained in this work correlate with other Heusler systems? Please discuss.”

Response: Thank you for your insightful comments. The zT value obtained in this study represents both an evolutionary step and a substantial advancement from traditional Heusler half-Heusler (HH) systems. Our approach builds upon an optimized ZrNiSn matrix, derived from existing literature (*Adv. Energy Mater.* **5**, 1500588 (2015)), where excess Ni reduces κ_L , and Sb doping optimizes n_H . This foundational strategy enabled our non-ALD coated samples to achieve a zT value of 1.1 at 873 K, one of the highest for n-type ZrNiSn systems.

Further, we employed ALD to introduce composite phase interface engineering. This method further simultaneously tunes the electron and phonon transport in ZrNiSn-based alloys, elevating the zT value to 1.3 at 873 K, the highest recorded to date in this context.

The precision of the ALD technique, which allows for precisely control over specific interfaces, is critical. This level of control, challenging to attain through traditional alloying and doping techniques, enhances the zT values significantly. Moreover, the application of coherent composite phase interface engineering via ALD, as developed in this work, not only benefits the ZrNiSn-based system but also holds promise for enhancing other HH and thermoelectric systems.

Detailed comment 6: “[6] Designating TiO₂ layers as “ $C = 0$, $C = 10$, $C = 20$, $C = 40$, $C = 60$ and $C = 80$ ” is misleading. If somebody wants to reproduce these results (perhaps by using another technique), more general descriptors are needed (e.g., thickness rather than cycles). For

example, C = 40 means that the TiO₂ thickness is about 3 nm (page 8). Why is this complicated designation (method oriented) necessary?”

Response: Thank you for your valuable comments on the nomenclature of the TiO₂ layer. Our initial rationale for using the number of ALD cycles as descriptors was to provide a direct and straightforward method to control and report the deposition of TiO₂ layers. We recognize that while specifying ALD cycles is simple and unambiguous for those familiar with the technique, it may not be universally applicable or intuitive for researchers using different deposition methods.

You rightly pointed out that using thickness as a descriptor could offer a more general and easily reproducible metric, though it is subject to variations based on the specific deposition technique, materials used, and process parameters. To bridge this gap, we have included detailed correlations between the number of ALD cycles and the actual thicknesses of TiO₂ layer in Table S6 of the revised supplementary materials. For instance, a designation of C = 40 corresponds to a TiO₂ layer approximately 3 nm thick.

We will heed your advice and implement a more standardized naming convention that emphasizes physical dimensions like thickness rather than process-specific metrics such as cycles. This adjustment will undoubtedly enhance the clarity, applicability, and reproducibility of our findings across different research settings.

Detailed comment 7: “[7] *It would be useful to designate the binding energy of both contributions in Fig. S3. Please also compare the binding energy with literature.*”

Response: Thank you for your valuable suggestion. We have marked the binding energies of Ti's 2p orbitals in the Fig. S3. Specifically, our fitting results indicate that the binding energies for the Ti 2p_{1/2} and Ti 2p_{3/2} orbitals are approximately 464.5 eV and 458.8 eV, respectively.

Furthermore, we have compared these values with the binding energy data from similar XPS studies reported in the literature on functional materials. Our findings align closely with these reports (*Nat. Commun.* **4**, 2214-2222 (2013), *Nat. Commun.* **11**, 4613-4621 (2020).) which consistently indicate a Ti valence state of +4 and corroborate the formation of amorphous TiO₂. We have included this comparative analysis in Fig. S3 in the revised Supplementary Information (SI) to provide a more comprehensive understanding and verification of our results.

Fig. S3. XPS spectra of Ti 2p_{1/2} and Ti 2p_{1/3} core level for TiO₂ = 3.2 nm sample.

Detailed comment 8: “[8] Does hydrogen have any role here? Having H₂O as a precursor should not be ignored. In many fields, H is instrumental. Please discuss.”

Response: Thank you for your question about the role of hydrogen in this study. The ALD cycle includes multiple steps: exposure to TDMAT, followed by an Ar (g) purge, exposure to H₂O vapor, and a final Ar (g) purge. The TDMAT precursor and the substrate are heated at temperatures of 40 °C and 150 °C, respectively, while the H₂O vapor is kept at room temperature.

During the H₂O exposure step, some of the water molecules react with the adsorbed TDMAT on the surface of the matrix particles. Any unreacted H₂O, along with by-products of the reaction, are then purged from the reaction chamber with argon gas. Additionally, the relatively low temperatures used in our process are designed to prevent the diffusion of elements, including hydrogen.

Based on the analysis and consistent with other studies using H₂O as an ALD precursor (*Nat. Commun.* **10**, 4166-4172, (2019), *Nanoscale* **3**, 1580-1588 (2020)), the presence of hydrogen in the deposited TiO₂ thin films and the final sample is negligible. This has been empirically confirmed

and is considered to have no significant impact on the properties of the films produced by the described ALD technique.

Detailed comment 9: “Fig.. S4 has never been mentioned in the manuscript. If it’s redundant, please remove it. Please check other Fig.s in the supplementary materials as well.”

Response: We apologize for the oversight of not mentioning Fig. S4 in the initial manuscript. It has been noted that Fig.. S2 and S4, while appearing similar, each depict critical stages of the experimental process. Fig.. S2 illustrates the XRD pattern of the powder post-ALD, highlighting an amorphous coating that is sufficiently thin—less than 10 nm—rendering it below the resolution capabilities of XRD for detecting secondary phase peaks.

In contrast, Fig. S4 represents the XRD diffraction pattern following SPS, which similarly does not display distinct secondary phase peaks. This absence is attributed to the composite phase content being below the detection threshold of XRD (5%). This phenomenon aligns with findings reported in related ALD studies, such as those documented in *Nano Energy*, **49**, 257–266, (2018).

An extensive review of all Fig.s in the supplementary materials was conducted, ensuring that each is accurately mentioned within the revised manuscript to clarify their relevance and importance to the study.

Detailed comment 10: “[10] Is the compositional analysis as accurate as presented in Table S1? Are all digits statistically reasonable to supply? What is the error bar? Why is O omitted in Table S1? Similar goes for Table S2.”

Response: We appreciate your attention to detail and constructive feedback. Here is a refined explanation addressing your concerns about the data presented in Tables S1 and S2:

1. Accuracy and error margin of EPMA analysis: EPMA is renowned for its high precision and quantitative accuracy, with a detection limit typically between 0.01 and 0.05 weight percent. The typical error margin for EPMA analysis is within $\pm 3\%$, ensuring that the presented data are within a reasonable uncertainty range. As a result, we assert that the compositional data in Tables S1 - S4 are both accurate and reliable.

2. Statistical validity: The EPMA point scanning analysis at points 1 and 2 indicates the Ti content at the Ti-ZNSS interface increases progressively with the number of ALD cycles. This trend substantiates the effective integration of Ti into the ZNSS matrix. At point 3, the matrix composition remains unaffected by the interface, displaying the original composition ($\text{ZrNi}_{1.03}\text{Sn}_{0.99}\text{Sb}_{0.01}$) without Ti. This consistency across multiple scans underscores the statistical robustness of our data.

3. Omission of oxygen: The focus of our point scanning data collection was primarily on the Ti-ZNSS layer and the matrix, not on the ZrO_2 secondary phase. Consequently, oxygen was omitted in Tables S1 to S4. However, the EPMA backscattered electron image in Fig. 3a of the main text clearly shows the presence of oxygen and confirms the formation of ZrO_2 .

Table S1. Point-scanned data in EPMA image of $\text{TiO}_2 = 0.8$ nm sample (1, 2 and 3 points).

position	Zr at%	Ni at%	Sn at%	Ti at%	Sb at%
1	30.375	33.755	32.895	2.602	0.373
2	30.672	33.304	32.693	2.969	0.362
3	32.157	34.237	33.221	0.000	0.381

Table S2. Point-scanned data in EPMA image of $\text{TiO}_2 = 1.6 \text{ nm}$ sample (1, 2, 3 points).

position	Zr at%	Ni at%	Sn at%	Ti at%	Sb at%
1	26.643	33.741	33.104	6.202	0.310
2	26.150	33.955	33.308	6.269	0.318
3	32.157	34.237	33.221	0.000	0.385

Table S3. Point-scanned data in EPMA image of $\text{TiO}_2 = 3.2 \text{ nm}$ sample (1, 2, 3 points).

position	Zr at%	Ni at%	Sn at%	Ti at%	Sb at%
1	23.770	33.141	32.928	9.874	0.287
2	23.526	33.315	33.897	9.969	0.293
3	33.154	33.554	32.971	0.000	0.321

Table S4. Point-scanned data in EPMA image of TiO₂ = 4.8 nm sample (1, 2, 3 points).

position	Zr at%	Ni at%	Sn at%	Ti at%	Sb at%
1	22.577	33.774	32.111	11.289	0.249
2	22.103	33.367	32.617	11.662	0.251
3	32.654	34.084	32.986	0.000	0.276

Detailed comment 11: “[11] The authors argue about strain in the interfaces. How is strain affecting the transport properties?”

Response: Thanks for your valuable comment. In thermoelectrics, strain distribution crucially influences both carrier and phonon transport. In particular, high-entropy alloys exhibit significant strain fields that intensify phonon scattering, effectively decreasing the κ_L , as corroborated by recent research (*Science* **384**, 81–86 (2024)).

Employing geometric phase analysis (GPA) technology, we precisely delineated local strain distributions. Contrary to traditional nanocomposites, where strain is predominantly localized at phase boundaries, our findings demonstrate a strain distribution throughout the entire particle. This strain distribution is particularly noteworthy along the ϵ_{xx} direction at the twin boundaries within the ZrO₂ structure, where it markedly impedes phonon propagation. This mechanism significantly contributes to the observed reduction in κ_L , highlighting the critical influence of strain on thermal transport in thermoelectric materials.

Detailed comment 12: “[12] Please provide a reference for the procedure pertaining to the Pisarenko curve.”

Response: We appreciate the reviewer's request for a reference detailing the procedure for the Pisarenko plot. For an in-depth description of this method, please refer to the study published in *Adv. Funct. Mater.* **23**, 5123–5130 (2013), which provides a comprehensive analysis of the Pisarenko curve and its applications in thermoelectric materials.

The mathematical formulation for the room-temperature m^* in the SPB model (*Phys. Rev. B.* **80**, 125205 (2009)) includes several components, with key equations provided below:

(1)

(2)

(3)

where e is the electron charge, k_B is the Boltzmann constant, \hbar is the reduced Planck constant, η is the reduced Fermi level and r is the scattering factor, which is taken as $r = -1/2$ for both the acoustic phonon scattering and alloy scattering, $F_j(\eta)$ is the j th order Fermi integral.

Detailed comment 13: “[13] There are some typos in the manuscript (e.g., “fromthe” on page 16 and “Mpa” on page 23 (it should be MPa)). Please proofread the text.”

Response: We appreciate the reviewer's attention to detail in identifying the typos in our manuscript. We sincerely apologize for these oversights and have thoroughly proofread the manuscript to correct these and any other errors.

Specifically, we have corrected "fromthe" to "from the" on page 16 and "Mpa" to "MPa" on page 23, ensuring all units and spacing are consistent with scientific standards.

Thank you for helping us improve the clarity and accuracy of our manuscript.

Detailed comment 14: “[14] Band structure in Fig.. S10 is given for ZrNiSn and TiNiSn as well as ZrO₂ in Fig.. S12, rather than the actual system investigated in the current work. Drawing conclusions from a very idealized model system(s) can be ambiguous. Please elaborate.”

Response: In view of the complex material composition studied herein, the reviewer raised a very important point. We address the reviewer’s question as follows:

Addressing the concern regarding the use of idealized model systems, it is important to highlight the inherent trade-offs between the reliability (accuracy and precision) and the cost of structural modeling. Computational studies frequently utilize idealized models due to the complexity and variability inherent in experimental setups. To mitigate any potential ambiguities arising from our modeling approach, we clarify that our coated samples are systematically designed and processed based on the matrix samples, with a specific focus on the thermoelectric performance influenced by the proposed composite phase interface engineering.

The composite phase interfaces are bifurcated into two main components: the Ti-ZNSS layer and the ZrO₂ nanoparticles. For the Ti-ZNSS layer, additional band structure calculations were conducted for Zr_{0.67}Ti_{0.33}NiSn. This is corroborated by EPMA point scanning data (Tables S1 - S4), which demonstrate a proportional increase in Ti content in the Ti-ZNSS layer as the thickness of the TiO₂ coating layer increases. This increase corresponds to an upward shift in the conduction band position as depicted in our band structure calculations (Fig.. S12, ZrNiSn-Zr_{0.67}Ti_{0.33}NiSn-TiNiSn). Considering that a minor excess of Ni is primarily employed to decrease lattice thermal conductivity, while a small amount of Sb doping is utilized to optimize carrier concentration, studies have reported that the influence of these modifications on the band structure is negligible (*Sci. Rep.* **4**, 6888 (2014), *ACS Appl. Mater. Interfaces* **11**, 47830–47836 (2019)). Thus, it is justifiable that band calculations omit the effects of both elements. Regarding ZrO₂, its robust chemical bonds significantly restrict the incorporation of additional elements into its matrix.

Furthermore, considering that the calculation of band structure is usually at 0 K, we have also conducted variable-temperature XRD tests, which show that the lattice constants of the ZrNi_{1.03}Sn_{0.99}Sb_{0.01} sample remain unchanged from room temperature to 873 K, indicating that the band structure calculated at the 0 K point is also applicable at high temperatures (Fig. S5).

Besides, the monoclinic phase of ZrO_2 also exhibits extremely small lattice temperature variations, which has been confirmed in the *Phys. Rev. B.* **70**, 245116 (2004).

Fig. S12. Band structure diagram of **a** ZrNiSn, TiNiSn and **b** $\text{Zr}_{0.67}\text{Ti}_{0.33}\text{NiSn}$.

Fig. S5. **a** Variable temperature XRD patterns of $\text{TiO}_2 = 3.2$ nm samples and **b** the corresponding lattice parameters

Detailed comment 15: “[15] The statement that the “base pressure of ~ 1 Torr” was achieved in ALD experiments doesn’t seem to be correct. This is probably the working pressure (1 Torr as a base pressure may give rise to full oxidation). Please state the base pressure (pressure before depositions).”

Response: We appreciate the reviewer's astute observation and apologize for any confusion regarding the pressure terminology in our ALD experiments. In this work, the TiO₂ layers were deposited on the matrix powder surface using a homemade continuous-flow ALD reactor. Ar gas was employed both as a carrier and a purging gas, with the working pressure maintained at approximately 1 Torr.

To clarify, the mentioned pressure of ~1 Torr refers to the working pressure during the deposition process, not the base pressure. The actual base pressure, measured before the start of the depositions, was maintained at approximately 5×10^{-6} Torr, ensuring a clean and controlled environment to prevent full oxidation.

Detailed comment 16: “[16] More details are needed for the characterization techniques (e.g., power setting for XRD, pass energy for XPS, accelerating voltage for microscopy, calibration procedure for DSC, etc.).”

Response: We appreciate the reviewer's request for additional details regarding our characterization techniques. To enhance transparency and ensure reproducibility, we have expanded the experimental section with the following specific details:

XRD testing conditions: For phase structure analysis using the Rigaku Smartlab 9kW X-ray diffractometer, we specified a tube voltage of 45 kV and a tube current of 200 mA.

XPS testing details: In our X-ray photoelectron spectroscopy (XPS) analysis, the pass energy was set to 20 eV, essential for accurate surface electronic state characterization.

EPMA and HRTEM parameters: The acceleration voltages for electron probe microanalysis (EPMA) and high-resolution transmission electron microscopy (HRTEM) ARM300 were set at 15 kV and 300 kV, respectively.

DSC calibration procedure: We detailed the calibration procedure for differential scanning calorimetry (DSC), using sapphire as a standard. This procedure includes blank tests, sapphire standard tests, and sample tests, ensuring precise baseline correction for each DSC curve.

REVIEWER COMMENTS

Reviewer #1 (Remarks to the Author):

The authors have adequately addressed all the comments that I raised. Therefore, I have no further comment, and in my opinion, it is ready for publication in Nature Communications.

Reviewer #2 (Remarks to the Author): attached

Reviewer #3 (Remarks to the Author):

The authors have improved the manuscript. The work is novel and of very high quality required for this journal. It should be provisionally accepted if the authors reduce the number of supplied digits in Table S1 – S4 to statistically relevant ones. In the rebuttal, they have claimed that “the typical error margin for EPMA analysis is within $\pm 3\%$ ”. Depending if this error is taken as absolute or relative, the Zr content in Table S1 (position 1) should be either 30 at.% or 30.4 at.% but certainly not 30.375 at.% (currently supplied value). For instance, reporting the composition with a “precision” higher than 1 at.% can often be very questionable (see, e.g., Surface & Coatings Technology 473 (2023) 130020). Surely the authors of the current manuscript don't have the precision of 0.001 at.% by EPMA.

The author answered and resolved the questions raised by the reviewer very carefully, and therefore agreed to publish.

Referee #3:

General comment: “The work is novel and of very high quality required for this journal. It should be provisionally accepted if the authors reduce the number of supplied digits in Table S1 – S4 to statistically relevant ones. In the rebuttal, they have claimed that “the typical error margin for EPMA analysis is within $\pm 3\%$ ”. Depending if this error is taken as absolute or relative, the Zr content in Table S1 (position 1) should be either 30 at.% or 30.4 at.% but certainly not 30.375 at.% (currently supplied value). For instance, reporting the composition with a “precision” higher than 1 at.% can often be very questionable (see, e.g., *Surface & Coatings Technology* 473 (2023) 130020). Surely the authors of the current manuscript don’t have the precision of 0.001 at.% by EPMA.”

General response: Thank you for your positive evaluation and the constructive comments on our manuscript. We have taken your suggestions about the precision of the EPMA data in Tables S1-S4 seriously and have made the necessary adjustments to our supplementary tables accordingly.

Table S1. Point-scanned data in EPMA image of TiO₂ = 0.8 nm sample (1, 2 and 3 points).

position	Zr at%	Ni at%	Sn at%	Ti at%	Sb at%
1	30.4	33.8	32.9	2.6	0.4
2	30.7	33.3	32.7	3.0	0.4
3	32.2	34.2	33.2	0.0	0.4

Table S2. Point-scanned data in EPMA image of $\text{TiO}_2 = 1.6$ nm sample (1, 2, 3 points).

position	Zr at%	Ni at%	Sn at%	Ti at%	Sb at%
1	26.6	33.7	33.1	6.2	0.3
2	26.2	34.0	33.3	6.3	0.3
3	32.2	34.2	33.2	0.0	0.4

Table S3. Point-scanned data in EPMA image of $\text{TiO}_2 = 3.2$ nm sample (1, 2, 3 points).

position	Zr at%	Ni at%	Sn at%	Ti at%	Sb at%
1	23.8	33.1	32.9	9.9	0.3
2	23.5	33.3	33.9	10.0	0.3
3	33.2	33.6	33.0	0.0	0.3

Table S4. Point-scanned data in EPMA image of TiO₂ = 4.8 nm sample (1, 2, 3 points).

position	Zr at%	Ni at%	Sn at%	Ti at%	Sb at%
1	22.6	33.8	32.1	11.3	0.2
2	22.1	33.4	32.6	11.7	0.3
3	32.7	34.1	33.0	0.0	0.3

REVIEWERS' COMMENTS

Reviewer #3 (Remarks to the Author):

The authors have modified the manuscript in a satisfactory manner. It's acceptable for publication in the current form.